# Deletion of NPAS4 in olfactory bulb principal neurons alters E/I balance and impairs decoding of chemically similar odour molecules

David Wolf[1,2] 🆔, Lars-Lennart Oettl[2] 🆔, Luis Sanchez-Guardado[3,4], Christiane Linster[5], Carlos Lois[3] and Wolfgang Kelsch[1,2] 🆔

[1] *Department of Psychiatry and Psychotherapy, University Medical Center, Johannes Gutenberg University, Mainz, Germany*

[2] *Central Institute of Mental Health, Medical Faculty Mannheim, Heidelberg University, Mannheim, Germany*

[3] *Division of Biology and Biological Engineering, California Institute of Technology, Pasadena, CA, USA*

[4] *Department of Cell Biology, School of Science, University of Extremadura, Badajoz, Spain*

[5] *Computational Physiology Laboratory, Department of Neurobiology and Behavior, Cornell University, Ithaca, New York, USA*

The peer review history is available in the Supporting Information section of this article (https://doi.org/10.1113/JP288011#support-information-section).

*The Journal of Physiology*

**Abstract figure legend** Conditional developmental deletion of the activity-dependent transcription factor NPAS4 in mitral and tufted cells ($\Delta$NPAS4$^{M/T}$) impacts coding in the adult main olfactory bulb (MOB). Schematic shows exemplary inhibitory interneurons (blue) that innervate MOB projection neurons (red) that then project to the anterior olfactory nucleus (AON). In vivo single-unit recordings in adult awake mice revealed that $\Delta$NPAS4$^{M/T}$ comes with reduced levels of odour-inhibited responses and temporally blurred odour-excited responses in these MOB projection neurons. This mutation still leaves odour decoding largely intact in MOB projection neurons, but impairs some coding features like that of chemical similarity. Downstream in the AON of $\Delta$NPAS4$^{M/T}$ mice, odour responses become more uniform and lose their robust decoding of different aldehydes. In summary, we identify NPAS4 as a factor in MOB projection neurons that is required for the formation of coordinated excitation-inhibition patterns and distinct representations to extract odour identity in a downstream cortex.

**Abstract** Odour representations are established in the olfactory bulb by a fine balance of excitatory and inhibitory activity. The projection neurons of the olfactory bulb, the mitral and tufted cells then pass this information to the olfactory cortices. While bulbar circuits have been studied at the neural and synaptic level, relatively little is known about the activity-dependent gene transcription machinery that shapes connectivity of mitral/tufted cells and thereby discriminative bulbar odour representations. As a first step, we conditionally deleted a candidate gene involved in synaptic wiring selectively in mitral and tufted cells during embryonic development and performed single-cell recordings in the olfactory bulb and the anterior olfactory nucleus of adult awake mice. We found that the activity-dependent transcription factor NPAS4 is necessary to establish temporally precise odour responses and normotypic levels of odour-inhibited responses in the adult olfactory bulb. The altered bulbar odour representations in NPAS4 mutants still contain information about odour identity, but show impaired coding of chemical similarity. Interestingly, odour responses in the cortex of NPAS4 mutants lose their robust decoding of different aldehydes. In summary, we identify NPAS4 as a factor in olfactory bulb projection neurons that is required for the formation of coordinated excitation-inhibition patterns and distinct representations of chemically similar stimuli to extract odour identity in the cortex. NPAS4 is part of a network of autism candidate genes. Considering this, these findings may contribute to a better understanding how alterations in synaptic wiring may contribute to the burden of neurodevelopmental disorders in the perceptual domain.

(Received 30 October 2024; accepted after revision 13 November 2025; first published online 5 December 2025)

**Corresponding author** W. Kelsch: Department of Psychiatry and Psychotherapy, University Medical Centre, Johannes Gutenberg University, Mainz, Germany.　　Email: wokelsch@uni-mainz.de

## Key points

- Chemically similar odour molecules generate different perceptual objects. The first processing station of odours, the olfactory bulb, produces highly discriminable representations of similar molecules for downstream cortical decoding. Inhibitory inputs to olfactory bulb projection neurons shape such discriminable representations.
- We tested how conditional deletion of the activity-dependent transcription factor NPAS4 in mitral/tufted cells affects bulbar and cortical odour representations.
- In single-unit recordings of adult awake mice, mutants had impaired odour-inhibited responses and blurred odour-excited responses in the bulb.
- While only decoding of chemical similarity was lost in the bulb, discriminative cortical decoding broke down more generally.
- These findings reveal functions of bulbar inhibition for cortical decoding and candidate mechanisms for autism-related genes in sensory dysfunction already in early sensory regions.

## Introduction

Olfactory sensory neurons sample information from odorants in the nasal cavities and convey this information to the main olfactory bulb (MOB). Coding in the MOB displays complex, but organized, inhibitory and excitatory responses to odorants and thereby produces highly discriminable representations even of similar molecules

**David Wolf** is a medical doctor and systems neuroscientist. At the Central Institute of Mental Health (Mannheim, Germany) and the University Medical Centre Mainz (Germany), he studied odour coding—how it is shaped by internal state and genetic factors—and the neural representations of social memory. He is currently a neurology resident and clinician–scientist at the Department of Epileptology, University Hospital Bonn (Germany) studying spatial coding using human single-neuron recordings.

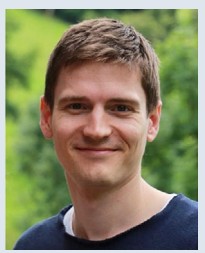

(Davison & Katz, 2007; Rinberg, 2006; Yokoi et al., 1995). The MOB directly broadcasts this information to a set of olfactory cortices. The most rostral olfactory cortex is the anterior olfactory nucleus (AON). The AON connects cortical hemispheres, provides feedback to the MOB (Brunjes et al., 2011) and is involved in perception, learning and state-dependent coding (Aqrabawi & Kim, 2020; Kikuta et al., 2008; Levinson et al., 2020; Oettl et al., 2016; Soria-Gómez et al., 2014; Wolf, Hartig et al., 2024). Cortical neurons broadly respond to different odours and differ in their patterns of odour-inhibited and -excited responses (Wolf, Oettl et al., 2024). Such coding is thought to emerge from activity-dependent neuronal wiring of excitatory and inhibitory synaptic connections (Kelsch et al., 2010; Sachse et al., 2007). Alterations in the neuronal wiring are therefore predicted to interfere with the decoding of olfactory information.

Mitral and tufted cells (hereafter collectively referred to as M/Ts) of the MOB are glutamatergic projection neurons that have one apical dendrite preferentially targeting one glomerulus, and multiple lateral dendrites. M/Ts typically project to multiple brain regions along the anterior-posterior ventral brain axis (Igarashi et al., 2012). During initial embryonic development, genetic factors govern axonal and dendritic outgrowth that is refined by activity-dependent wiring (Imai, 2014; Mombaerts, 2001; Ravi et al., 2017). Relatively little is known about the genes that are responsible for the activity-dependent synaptic wiring and synapse maturation in M/Ts (Ravi et al., 2017) with their potential impact on odour coding in the MOB and the downstream cortices in adult animals.

Here, we tested the consequences of a developmental gene deletion in M/Ts of a transcription factor that preferentially affects the maturation of inhibition in the neocortex. The activity-dependent transcription factor Neuronal PAS Domain Protein 4 (NPAS4) is required in cortical principal neurons to produce a balance between excitatory and inhibitory synapses (Fu et al., 2020). NPAS4 is responsive to excitation-coupled postsynaptic calcium influx (Bloodgood et al., 2013). While the function of NPAS4 is not known for M/T maturation and their odour coding in the adult (Fujimoto et al., 2023; Ravi et al., 2017; Yoshihara et al., 2014), NPAS4 activates distinct programs of late-response genes in inhibitory and excitatory neocortical neurons (Lin et al., 2008; Spiegel et al., 2014). NPAS4 is rapidly induced in glutamatergic neurons in response to activity-dependent calcium influx (Bloodgood et al., 2013; Lin et al., 2008). Once expressed, NPAS4 regulates the transcription of specific downstream genes that promote the formation, maintenance, and function of inhibitory synapses onto these excitatory neurons (Lin et al., 2008). NPAS4 does not increase inhibition globally; it specifically enhances inhibitory input onto active excitatory neurons, allowing for precise homeostatic control for instance through compartment-specific BDNF expression (Bloodgood et al., 2013; Spiegel et al., 2014). Accordingly, we examined how developmental deletion of NPAS4 in M/Ts affects odour-evoked representations in adult mice.

Towards this aim, we generated a mutant mouse strain with selective deletion of the gene coding for NPAS4 in M/Ts. We characterized the features of odour responses in the MOB and AON of adult mutants. Towards this aim, we performed recordings of populations of single-units in awake mice. Mice were presented with a set of molecularly similar aldehyde odorants with increasing carbon chain length (Imamura et al., 1992; Mori et al., 1992; Sato et al., 1994; Yokoi et al., 1995). These molecularly similar odorants are suited to address discriminative coding and the role of inhibition: behavioural data from a number of species (including rats, Linster & Hasselmo, 1999; squirrel monkeys, Laska & Freyer, 1997; Laska & Teubner, 1998, 1999; and honeybees, Laska et al., 1999) support the utility of the straight-chain aliphatic homologous series as an axis for graded odorant similarity. Greater overlaps between odorants' activation patterns should correlate with an increased perceptual similarity of those odorants (Cleland et al., 2002; Yoshida & Mori, 2007). These aldehydes preferentially activate subsets of dorsomedial glomeruli (Imamura et al., 1992; Mori et al., 1992; Sato et al., 1994). At the glomerular input level of the olfactory bulb, structurally related molecules elicit activation in overlapping sets of glomeruli (Johnson & Leon, 2000; Rubin & Katz, 1999; Wachowiak & Cohen, 2001). At the level of bulbar output, certain aldehydes excite a M/T, while another closely related aldehyde inhibits the same M/T (Yokoi et al., 1995), an effect that emerges from lateral inhibition among M/T via dendro-dendritic inhibition of granule cell interneurons (Arevian et al., 2008; Margrie et al., 2001; Yokoi et al., 1995). Perceptual discrimination between these odorants can be modulated via changes in excitation and inhibition (Abraham et al., 2010; Chaudhury et al., 2009; Mandairon et al., 2006). Indeed, we find that deletion of NPAS4 in M/Ts results in altered odour response features and impaired encoding of chemical similarity in the MOB and odour identity in the AON of adult mice.

## Materials and methods

### Ethical approval

All procedures were approved by the local animal welfare authority (Regierungspräsidium Karlsruhe, ethics committee reference number 35-9181/G-229/14) and performed in accordance with the EU Directive 2010/63 and The Journal of Physiology's policies regarding animal experiments. All efforts were made to minimize pain and suffering. The investigators acknowledge the animal ethical principles under which the journal operates and

confirm the compliance of the present work. Inhalation anaesthesia was used for surgery, intraperitoneal injection for terminal procedures as stated in the respective sections.

## Animals and husbandry

To examine the recombination pattern of Tbet-Cre-positive mice (B6;CBA-Tg(Tbx21-cre)1Dlc/J, RRID:IMSR_JAX:02 4507, JAX stock #02 4507, Jackson labs, Haddad et al., 2013), Tbet-Cre-positive mice were crossed with Ai9 reporter mice (B6.Cg-*Gt(ROSA)26Sor*$^{tm9(CAG-tdTomato)Hze}$/J, RRID:IMSR_JAX:0 07909, JAX stock #0 07909, Jackson labs), producing Tbet-Cre x Ai9 mice. To obtain mice with a developmental deletion of the genes encoding for NPAS4 selectively in M/Ts, Tbet-Cre-positive mice (B6;CBA-Tg(Tbx21-cre)1Dlc/J, RRID:IMSR_JAX:02 4507, JAX stock #02 4507, Jackson labs, Haddad et al., 2013) were crossed with NPAS4$_{fl/fl}$ mice (Npas4$^{tm1Meg}$, RRID:MGI:3 828 101, gift of Dr. M.E. Greenberg, Lin et al., 2008), producing $\Delta$NPAS4$^{M/T}$ (Cre+ x NPAS4$_{fl/fl}$) or control mice (Cre– x NPAS4$_{fl/fl}$). All transgenic mice were bred in-house and maintained in a C57BL/6J (Charles River Laboratories) background (>F10). For MOB recordings, we used five $\Delta$NPAS4$^{M/T}$ and five control mice. For AON recordings, we used six $\Delta$NPAS4$^{M/T}$ and nine control mice. Animals were single housed following surgical procedures, supplied with ad-libitum access to food and water for the complete duration of the experiments and kept on a 12-h light-dark-cycle. The unit recordings from the control mice were also used in another study performing site-by-site comparisons of olfactory cortices to address the question of region-specific coding in cortices (Wolf, Oettl et al., 2024).

## Immunohistochemistry

Mice were anesthetized with ketamine/xylazine (300 mg/kg BW ketamine and 60 mg/kg BW xylazine diluted in 0,9% saline, I.P. injection). At postnatal day 0 (P0), mouse brains were fixed by immersion using 4% paraformaldehyde (PFA) in 0.1M phosphate-buffered saline (PBS, pH 7.4) at 4°C overnight. Adult mice (P60) were fixed by intracardiac perfusion with 4% PFA in PBS, and then the brains were removed and maintained in 4% PFA at 4°C overnight. The next day, brains were rinsed with PBS and cryoprotected with 10% sucrose in PBS at 4°C overnight. Later, brains were embedded in 3% gelatin, frozen, and sectioned with a cryostat into 20 μm thick sections (Sánchez-Guardado & Lois, 2019).

Immunohistochemistry was performed in mouse brain sections using antibodies against the NPAS4 protein and Tbr2 protein. M/Ts and periglomerular cells express Tbr2 from embryonic to adult stages (Mizuguchi et al., 2012). Sections were incubated for 60 min in a blocking solution containing 1% bovine serum albumin (BSA) in PBS with 0.1% Triton X-100 (PBS-T). Then, sections were incubated overnight at room temperature with the primary antibodies against NPAS4 protein (1:5,000, kind gift from Dr. M.E. Greenberg, Lin et al., 2008) and Tbr2 protein (1:500 rat anti-Tbr2, Thermo Fisher Scientific, RRID:AB_11 042 577) diluted into blocking solution. The next day, sections were rinsed three times with PBS-T, 10 min each, followed by incubation with secondary antibodies goat anti-rabbit IgG Alexa-555 (RRID:AB_2 535 850) and donkey anti-rat IgG Alexa-488 (RRID:AB_2 535 794) diluted 1:1,000 in blocking solution. Finally, the sections were mounted with Fluoromount (Sigma-Aldrich). Selected sections we dipped into DAPI (4′,6-diamidino-2-phenylindole, dilution of 1:10 000, Merck) for nuclear counterstains.

Z-stacks images were acquired using 10x, 20x, or 40x objectives on a confocal microscope (Zeiss LSM 800). Z-stacks were merged and analysed using ImageJ and edited with Photoshop (Adobe) software. Overview pictures were acquired as montages.

## Odour delivery

For odour delivery, we used a custom-built air-dilution olfactometer (Shusterman et al., 2011). Odorants were diluted to a concentration of 1% in mineral oil and stored in dark vials flushed with nitrogen to prevent oxidation. Odorants were further diluted 1:10 in a constant air stream, resulting in a final odour concentration of 0.1%. We used the following aldehyde odorants of increasing carbon chain length: Propanal (538 124, Sigma Aldrich), Butanal (418 102, Sigma Aldrich), Pentanal (110 132, Sigma Aldrich), Hexanal (115 606, Sigma Aldrich), Heptanal (H2120, Sigma Aldrich), Octanal (O5608, Sigma Aldrich). In each trial, a single odorant was presented to the experimental subject in head-fixed configuration for 500 ms. Different odorants were presented in random order. To obtain balanced statistics, we included the first 19 trials of each odour for further analysis (which is the minimum number of trials for each aldehyde). The olfactometer valves were controlled using micro-controllers (Arduino, RRID:SCR_02 4884), connected to a MATLAB (Mathworks, RRID:SCR_0 01622) script to control the trial structure.

## In-vivo electrophysiology

MOB recordings were performed using silicon probes in tetrode configuration (A4x2-tet-5mm-150-200-121, NeuroNexus) attached to a custom-designed titanium

microdrive (Oettl et al., 2020). AON recordings were performed using custom-built tetrode arrays.

For implantation of recording arrays, mice underwent anaesthesia using isoflurane (1%–3% in 100% O2), and pre- and post-surgery analgesia (2 mg/kg BW meloxicam s.c. injection, Metacam Boehringer Ingelheim) was administered. During surgery, sufficient anaesthesia depth was regularly determined by testing pedal withdrawal reflex and monitoring of respiration rate and pattern. Once anesthetized, the mice were positioned in a stereotactic instrument with non-traumatic ear bars (Kopf Instruments) and placed on a warming pad (Stoelting Rodent Warmer). Topical lidocaine was applied. The dorsal skull's fur was removed, and the exposed skin was disinfected. Subsequently, the skin covering the dorsal cranium was resected, and the resection margins were fixed with tissue adhesive (3M Vetbond). Then the skull was prepared for craniotomy, which was performed centred (in mm) at 4.0 anterior, 1.0 lateral in relation to Bregma for MOB implants and at 3.0 anterior, 0.95 lateral for AON implants. Additionally, a hole was drilled in the posterior skull for implantation of the ground pin (Gold pin, NeuraLynx). Dental multi-component cement (C&B Superbond, Sun Medical) was applied to the skull, excluding the craniotomy holes. The recording array was prepared and attached to a motorized 3-axis micromanipulator (Luigs & Neumann) and lowered to the target position. Target depth for AON implants was 3.5 mm ventral from the brain surface. MOB drives were lowered until the base plate touched the dorsal skull surface, and the tips of the silicon probe were right above the dorsal MOB. The silicon probe and movable parts of the drive were shielded from the cement by covering them in a mixture of bone wax and mineral oil. Dental cement (Paladur, Kulzer) was then used to create a mechanical connection between the implant and the Superbond-covered skull. Post-surgery, mice were closely monitored and allowed to recover in their home cage for at least seven days before progressing to habituation for head-fixed experiments. Upon start of the recording period MOB drives were lowered until units with M/T features such as firing in rhythm with respiration, expected baseline firing rates (Fig. 2*C1*) and spike waveforms (Fig. 2*C2*) were detected in the anatomical depth expected for the mitral cell layer, consistent with previously described M/T features (Kay & Laurent, 1999; Kollo et al., 2014; Rinberg, 2006; Shusterman et al., 2011). To obtain stable units, recordings were performed 24 h after moving the recording shank.

### Histological confirmation of recording site

For histological confirmation of recording sites in the AON after experiment completion, animals were anesthetized with ketamine/xylazine (300 mg/kg BW ketamine and 60 mg/kg BW xylazine diluted in 0,9% saline, I.P. injection), followed by transcardial perfusion with 0.9% PBS and then 4% paraformaldehyde for fixation. Further immersion fixation in 4% paraformaldehyde of the whole skull for 2 weeks allowed visualization of the tetrode tracts. Serial coronal sections were prepared using a vibratome with 200 μm slices for visualization of tetrode tracts. Images were acquired on a Axio Imager 2 (Zeiss).

### Data acquisition and processing

Electrophysiological recordings from head-fixed mice were acquired in a Faraday cage using the RHD2000 Interface board (Intan Technologies) and 32- or 64-channel miniature amplifier headstages. Odour onsets and offsets were recorded as digital inputs from the olfactometer. Data were sampled at 30 kHz. Raw data was converted into MATLAB format for further processing and spike sorting which was conducted using in-house routines (Oettl et al., 2020; Winkelmeier et al., 2022). In brief, after common average referencing and filtering (300–5000 Hz, $4^{th}$ order Butterworth filter) of the raw traces, putative spike events were detected based on an amplitude threshold (exceeding $7.5 \times$ the median deviation of the absolute values of the pre-processed signal). Putative spike events were then clustered into single-units using a graphical user interface-based tool developed by A. Koulakov (CSHL) where single-units were identified based on waveform features (e.g. amplitude or principal component analysis of the waveforms on neighbouring channels). Single-unit quality metrics were computed using the mlib toolbox by M. Stüttgen (Version 6, https://de.mathworks.com/matlabcentral/fileexchange/37339-mlib-toolbox-for-analyzingspike-data) and the UltraMegaSort2000 toolbox (Hill et al., 2011) and assessed for noise contamination and stability over time. In the AON, we included units with a mean firing rate above 0.5 Hz.

A total of 167 units in the MOB ($n = 71$ units from 18 sessions in five control mice and $n = 96$ units from 18 sessions in five $\Delta$NPAS4$^{M/T}$ mice) and a total of 268 units in the AON ($n = 108$ units from 16 sessions in nine control mice and $n = 160$ units from 11 sessions in six $\Delta$NPAS4$^{M/T}$ mice) were included in the analysis. The units in control mice were used also for other questions and analyses in another study (Wolf, Oettl et al., 2024).

### Analysis of single-unit activity

Single-unit responses to the odours were first analysed based on their firing rate change. After alignment of trials to odour onset, peri-stimulus time histograms (PSTH) were computed with a bin width of 50 ms. To

assess the firing rate change in relation to the baseline firing rate and variability, we computed the z-scored response for every unit. Responses to the same odour were averaged across trials and the z-score of a unit's activity was computed at every time bin (100 ms bin width) $t$ by: $z_t = \frac{FR_t - FR_{baseline}}{\sigma_{baseline}}$, where $FR_{baseline}$ is the mean firing rate during the baseline window ($-2.5$ to $-0.5$ s) and $\sigma_{baseline}$ is the standard deviation of the firing rate during the baseline window. Units were classified as odour-excited if the mean z-scored response during the odour presentation (0–0.5 s relative to odour onset) exceeded $+1.96$ (which corresponds to two standard deviations of the firing rate distribution during the baseline window) and classified as odour-inhibited if the mean z-scored responses dropped below $-1.96$. The fraction of units responsive to any odour, and the odour-excited and -inhibited cell-odour pairs were visualized as pie charts. Next, for all odour-excited and odour-inhibited responses separately, we determined the amplitude of the z-scored response, the response peak width and the time to peak response (using the MATLAB built-in function 'findpeaks'). Statistical comparisons between responses from control or mutant animals were conducted in PRISM 9 (GraphPad Software). Normality of the distributions was tested with the Anderson-Darling and Shapiro–Wilk tests. Normally distributed data were tested using two-tailed two-sample $t$ tests, distributions outside the normal family were tested using two-tailed Mann–Whitney $U$-tests. Exact $P$-values are indicated in the respective figure panel and sample size and test statistic are indicated in the figure legends or corresponding results section. As the same animal was recorded more than once, we removed all M/T units that occurred on the same tetrode in subsequent sessions. Similar results were obtained replicating the same statistical differences as in Fig. 3 (data not shown), thereby ruling out the potential confounds originating from putative resampling.

To assess if the gene deletion in M/Ts had an influence on the differentiation of chemical similarity, we compared the range of odour responses as a function of chemical similarity in single-units (Chaudhury et al., 2009). For every unit, we identified the strongest odour response and normalized the responses to the remaining odours by it. We then sorted the relative response strengths by chemical distance (difference of carbon chain length of the aldehyde odorants). We plotted the average across responses from all single-units as a function of the carbon chain length separately for the two regions and genotypes and compared the mutants to controls statistically using an ordinary two-way ANOVA (with the factors 'mutant' and 'difference in carbon chain length'). A similar approach was used to compare the overall differentiation of odours by the range of odour responses. Again, the strongest odour response of a single-unit was used for normalization. Then odour

responses were ranked by strength. We plotted the average across responses from all single-unit as a function of the ranked responses and compared the mutants to controls using a repeated-measures two-way ANOVA (matching the responses of an individual single-unit across ranks; factors 'mutant' and 'response rank').

To quantify the discriminability of odour responses, we compared for every unit all discriminability indices for pairwise responses: $d' = \frac{|\mu_A - \mu_B|}{\sigma_{RMS}}$, where $\mu$ denotes the spike count during the stimulus presentation window (0 to $+0.5$ s relative to odour onset) across trials and $\sigma_{RMS}$ denotes the root-mean-square value of the standard deviations of spike counts across trials. The resulting distributions of average and maximum discriminability indices per unit were compared between mutant and control mice using a two-sided Wilcoxon ranksum test.

### Cross-odour distance

To quantify the difference in odour responses in the full population of recorded units, we computed the Euclidean distances between the population responses. First, we constructed the mean population response of every trial by computing the mean spike count during the odour presentation (0 to $+0.5$ s relative to odour onset). The resulting population vector, which has the dimensionality of the number of recorded units in that population, captures the population's response to the odour in that trial. We then computed the Euclidean distances between the population vectors of two trials, considering all possible trial combinations. The comparison of the distribution of distances between trials of a given odour (i.e. the trial-to-trial variability of the odour response) and the distribution of distances between the given and another odour informs us on the discriminability of the two odour population responses. We statistically tested the cross-odour distances against the within-odour trial-to-trial variability using the Brown-Forsythe ANOVA with Games-Howell's multiple comparisons test.

### Odour response classifier

Based on the results on the cross-odour distances, we tested if odour population responses in mutants decreased odour discriminability by measuring and comparing the accuracy of an odour response classifier (Oettl et al., 2020). To account for different sizes of the recorded populations, we used subsampling and compared the classifier accuracy between mutants and controls under equal conditions. Separately for populations from different regions or genotypes, we repeatedly ($n = 500$ iterations per step) drew random subsamples of the whole population, increasing the number of units included in steps of 10

until reaching the maximum number of recorded units in a population. At each iteration, first the population response vectors of all odours were jointly reduced to a dimensionality of four using PCA. This avoids over-fitting, since it reduces the number of parameters of the multivariate Gaussians that have to be estimated from the training data (Oettl et al., 2020). Then, a Gaussian mixture model was fit to the training data. The accuracy was tested using a leave-one-trial-out procedure, where one random trial was removed from the training dataset for testing. The total accuracy at each step was given as the fraction of correct predictions. To test if the decoding accuracy exceeded the chance level, we used a one-sided binomial test (chance level $= 0.1667$, $\alpha = 0.05$). To directly compare the discrimination of odour representations between mutants and controls, we analysed the classifier accuracies using generalized linear mixed effect models. For both regions separately, we employed a generalized linear mixed-effects model with a binomial link function to predict the outcome of the classifier prediction from the genotype. To account for the hierarchical structure of the data due to the repeated measurements (iterations) within each number of units drawn from the full population, we included random effects in the models. The number of units (grouping factor) was considered as random slopes and intercepts. Tukey's post-hoc test followed by Bonferroni adjustment for multiple comparisons was used to compare the mutant's effect on decoding accuracy. Models and statistical tests were conducted in R (https://www.r-project.org/) using the 'lme4', 'lmerTest' and 'multcomp' packages. To rule out that the observed loss of decoding accuracy in $\triangle NPAS4^{M/T}$ mutants in the AON depended on the specific decoding model chosen, we repeated the analysis as described above using multi-class error-correcting output codes (ECOC) models using binary $k$-nearest neighbours learners (Matlab function 'fitcecoc'). This control analysis reproduced the same results obtained with Gaussian Mixture Models.

## Statistical analyses

Data analysis was performed in MATLAB (MathWorks). Outputs were exported and statistical analyses were performed in Prism 9 (GraphPad Software). Source data for all statistical tests are provided in the Supporting Information for online publication. Exact p-values are indicated to three significant figures, test procedures are described in the respective methods section and test details are indicated in the figure legends or results section. For tests with many comparisons, a p-value table is additionally provided in the Supporting Information for online publication. Custom-written code and source data to reproduce all presented findings can be obtained from the lead author upon reasonable request.

## Results

We first report results on odour coding in M/T cells in the MOB of mice with deletion of NPAS4 in M/Ts and matched control mice. A detailed analysis of single odour responses is followed by an analysis of how odours of varying similarity are encoded at the population level. We then perform the single-unit and population analyses in the AON to test how modulation of bulbar output affects downstream odour representations.

### NPAS4 expression in M/T cells in the and MOB and its selective deletion

Tbr2 is a cell type marker for M/Ts in the MOB that is present at birth (Fig. 1*A*) and in the adult (Fig. 1*B*) (Mizuguchi et al., 2012). Also, NPAS4 is expressed in the granule, mitral and periglomerular layer of the MOB (Fig. 1*A,B*). As expected for a gene with activity-dependent induction, NPAS4 protein expresses under baseline conditions in a fraction of Tbr2-immunoreactive cells in the mitral cell layer and also in the external plexiform and periglomerular layer at a snapshot at postnatal day 0 or in adult mice (Fig. 1*A*–*C*).

The gene of interest was deleted conditionally by Cre-recombination under the Tbet promoter (also known as *Tbx21*, Haddad et al., 2013) that is expressed in M/Ts approximately from embryonic day 14 on (Faedo et al., 2002; Mizuguchi et al., 2012). We examined Tbet-Cre recombination in tdTomato reporter mice in coronal sections of adult mice (Fig. 1*D*). The Tbet promoter showed Cre-dependent recombination followed by tdTomato expression mainly in M/Ts in the MOB (Fig. 1*E1*). At higher magnification, tdTomato expression was also observed in some putative external tufted cells (Fig. 1*E2*, 1*E3*), consistent with prior reports (Kosaka & Kosaka, 2012). The tdTomato expression was also observed in the lateral olfactory tract where axons of M/Ts are densely packed (Fig. 1*F1*). In the AON, only axons and their terminal buttons expressed tdTomato (Fig. 1*F2*), consistent with expression of Tbet-Cre in the MOB, but not AON. We thus used Tbet-Cre mediated NPAS deletion in mice carrying two floxed NPAS4 alleles to obtain $\triangle NPAS4^{M/T}$ (Cre+ x NPAS4$_{fl/fl}$), or control mice (Cre- x NPAS4$_{fl/f}$).

### NPAS4-deficiency modifies odour responses in M/Ts

We recorded from male Tbet-Cre- and Tbet-Cre+ NPAS4$_{fl/fl}$ mice backcrossed >F10 to C57BL/6J mice that were fully adult (>P90). During awake head-fixed recordings, we applied six aldehyde odorants of increasing carbon chain length, $C_3H_6O$ to $C_8H_{16}O$ (Fig. 2*A*). The approximate position of the silicone probe in tetrode

configuration is illustrated in Fig. 2*B* targeting the dorsal MOB. Drives were lowered until units with M/T features such as firing in rhythm with respiration, expected baseline firing rates (Fig. 2*C1*) and spike waveforms (Fig. 2*C2*) were detected in the anatomical depth expected for the mitral cell layer (Kay & Laurent, 1999; Kollo et al., 2014; Rinberg, 2006; Shusterman et al., 2011). To obtain stable units, recordings were performed 24 h after moving the recording shank. Single-units had comparable spike waveforms and baseline firing rates in the MOB across genotypes (Fig. 2*C*, $n = 71$ units for control and $n = 96$ units for $\triangle$NPAS4$^{M/MT}$). The heatmaps (Fig. 2*D*) show average responses of all recorded M/Ts in $\triangle$NPAS4$^{M/T}$ mice to the series of aldehydes. For each odour, units are sorted according to their average *z*-scored rate response. Z-score responses of the five units with the highest average pairwise discriminability index (i.e. consistently different

odour responses; for details see Method section '*Analysis of single-unit activity*') from control (Fig. 2*E*) and mutant mice (Fig. 2*F*) show the different rate response patterns to the aldehydes of increasing c-chain length as previously described in (Yokoi et al., 1995).

Compared to *control* mice, adult $\triangle$NPAS4$^{M/T}$ mice had a smaller fraction of putative M/Ts that responded to at least one of the aldehyde odours (Fig. 3*A*). M/Ts in mutants displayed a more prominent loss of odour-inhibited than -excited responses (Fig. 3*B*). We then analysed the remaining odour-excited responses in mutants and control mice (Fig. 3*C*, $n = 36$ unit-odour pairs for control and $n = 37$ unit-odour pairs for $\triangle$NPAS4$^{M/T}$). Figure 3*D* shows the average odour z-scores of excited responses in control and mutant mice; note the different average shape of the response between genotypes. Specifically, M/Ts in mutants had

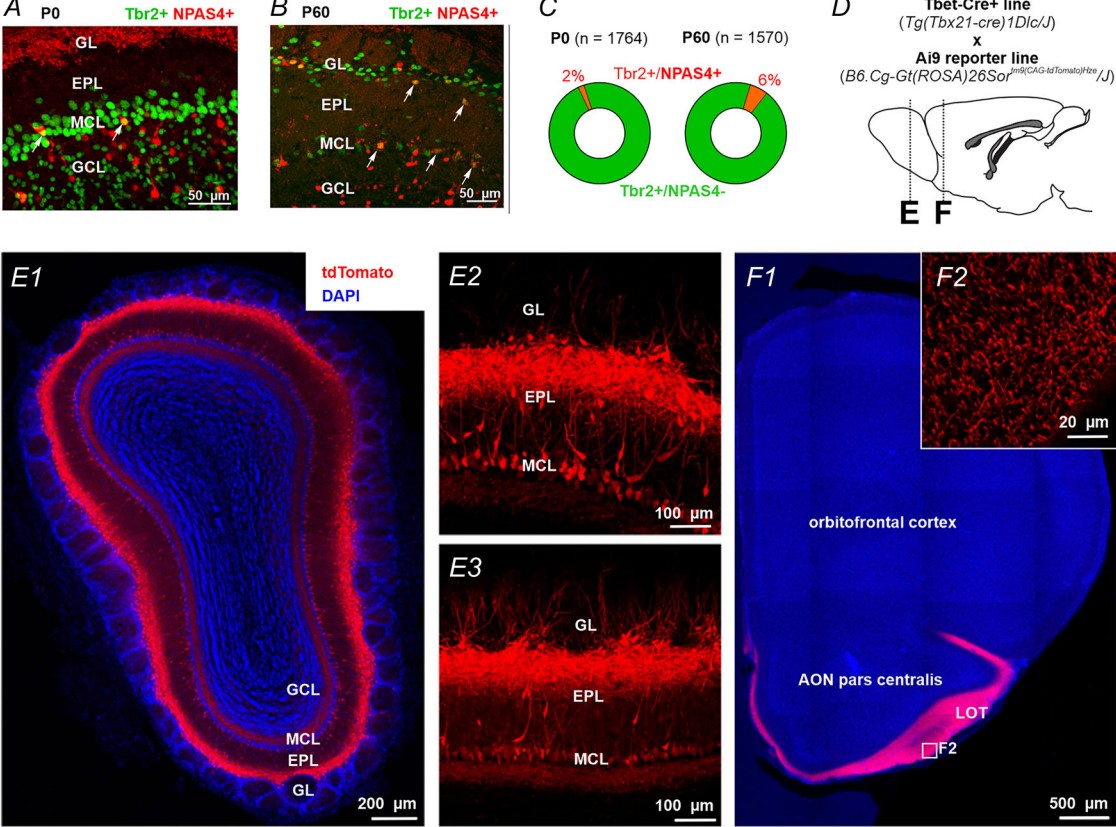

**Figure 1. NPAS in the MOB and selective recombination in Tbet-Cre mice in the anterior olfactory system.**
*A*, Exemplary MOB section at postnatal day 0 in control mice stained for Tbr2 and NPAS4 showing colocalization in a small fraction of cells in the mitral cell layer (MCL). External plexiform layer (EPL), granule cell layer (GCL), glomerular layer (GL). Scale bar, 50 μm. *B*, MOB section of an adult control mouse stained for Tbr2 and NPAS4 showing colocalization. Again, a small fraction of cells colocalized in the MCL. Scale bar, 50 μm. *C*, Fraction of cells immuno-reactive to both Tbr2 and NPAS4, or only Tbr2 in perinatal and adult control mice in the mitral cell layer. *D*, To initially examine the recombination pattern of Tbet-Cre mice, Tbet-Cre-positive mice were crossed with Ai9 reporter mice. *E*, Recombination pattern in the MOB of P60 mice in a coronal overview section with DAPI counterstain (E1) and two images at higher resolution (E2, E3). Recombination is restricted to mitral and tufted cells. *F*, In the AON, the LOT is densely labelled in the overview section with DAPI counterstain (F1). The inset (F2) taken from F1 reveals dTomato-positive axons and their buttons in Tbet-Cre x Ai9 mice at P60.

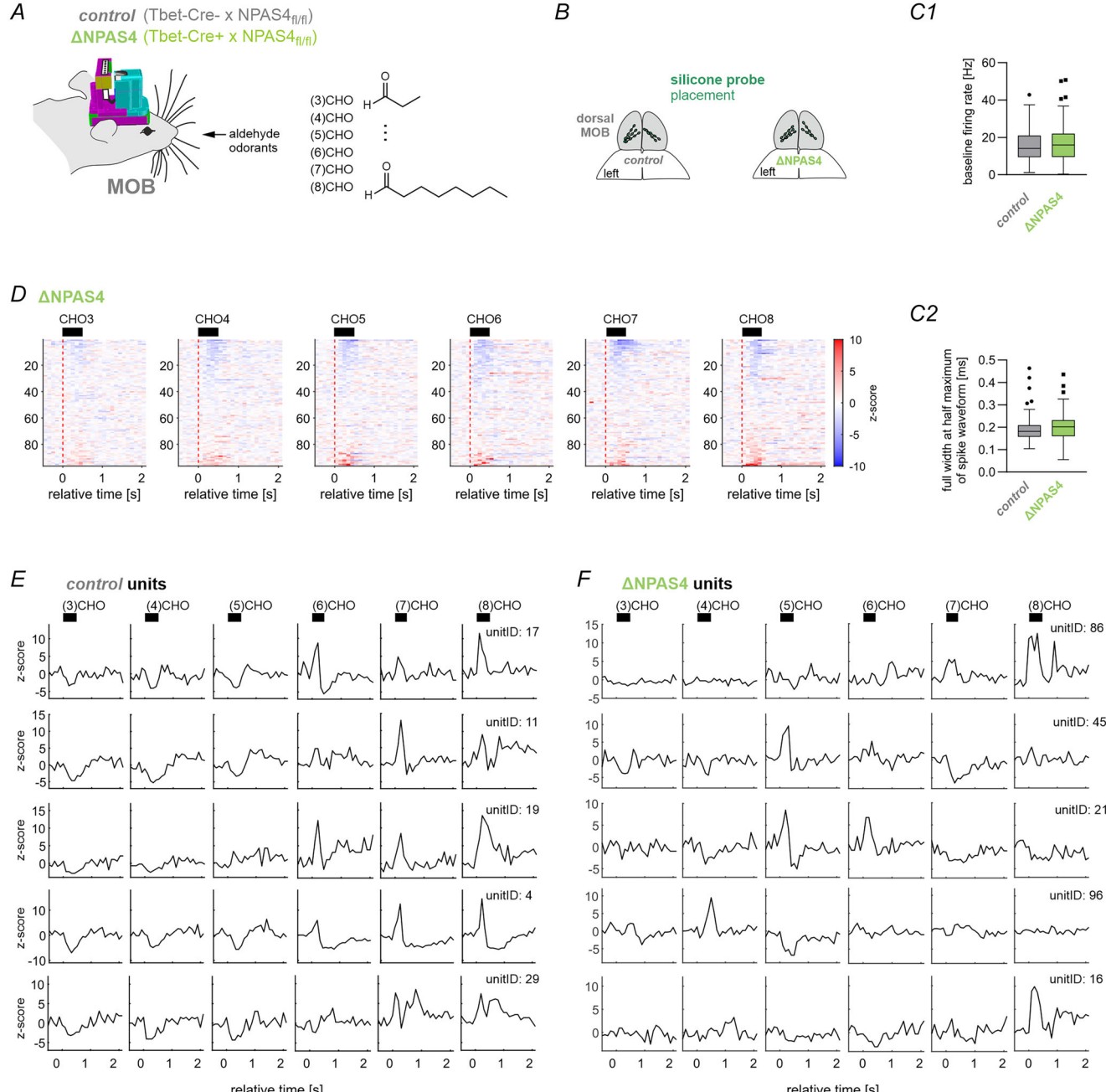

**Figure 2. Odour responses to aldehyde odorants in the MOB.**
*A*, ΔNPAS4$^{M/T}$ and control mice with chronically implanted silicon probes for single-unit recordings were placed in a head-fixed recording setup and presented aldehyde odorants with increasing lengths of the carbon chain (Propanal, (3)CHO, to Octanal, (8)CHO). In every trial, one of the odors was presented for 500 ms in pseudorandomized order. *B*, Anatomical positioning of the silicon probes used for MOB recordings. *C*, The baseline firing rate (**C1**) and the median full-width at half maximum of spike waveforms (**C2**) of recorded units in the MOB of the mutant or control mice ($n = 71$ units for control and $n = 96$ units for ΔNPAS4$^{M/MT}$). *D*, Trial-averaged z-scored responses of every unit from MOB to the different odour types in ΔNPAS4$^{M/MT}$ mice ($n = 96$ units). For each odour, the units were sorted based on their mean z-score during 0 to +0.5 s relative to odour onset. For matching response profiles in control mice see Wolf, Oettl et al. (2024). *E–F*, The z-scored responses to the different odors of the five responsive units with the highest average discriminability index d' from control (*E*) and ΔNPAS4$^{M/MT}$ (*F*). In this figure: Boxes show median and inter-quartile distance; whiskers extend to the most extreme points within 1.5×IQR; points beyond are plotted individually.

smaller amplitudes of odour responses than *control* mice (U = 402, P = 0.0033, Mann-Whitney (MW) U-Test, Fig. 3*E*). Mutants also had broader responses (U = 481.5, P = 0.0415, MWU-Test, Fig. 3*F*), and their responses peaked slightly later (U = 341.5, P = 0.0002, MWU-Test, Fig. 3*G*). We then examined the features of the odour-inhibited responses to these monomolecular odorants in the two genotypes (Fig. 3*H*, n = 56 unit-odour pairs for control and n = 31 unit-odour pairs for

$\Delta$NPAS4$^{M/T}$). The average response across M/Ts had a similar shape between genotypes (Fig. 3*I*). Specifically, when comparing the genotypes, the odour-inhibited responses had similar amplitudes (U = 767, P = 0.375, MWU-Test, Fig. 3*J*), similar peak width (t(85) = 0.3, P = 0.797, unpaired *t* test, Fig. 3*K*), but differed slightly in time to peak response (U = 630.5, P = 0.0244, MWU-test, Fig. 3*L*).

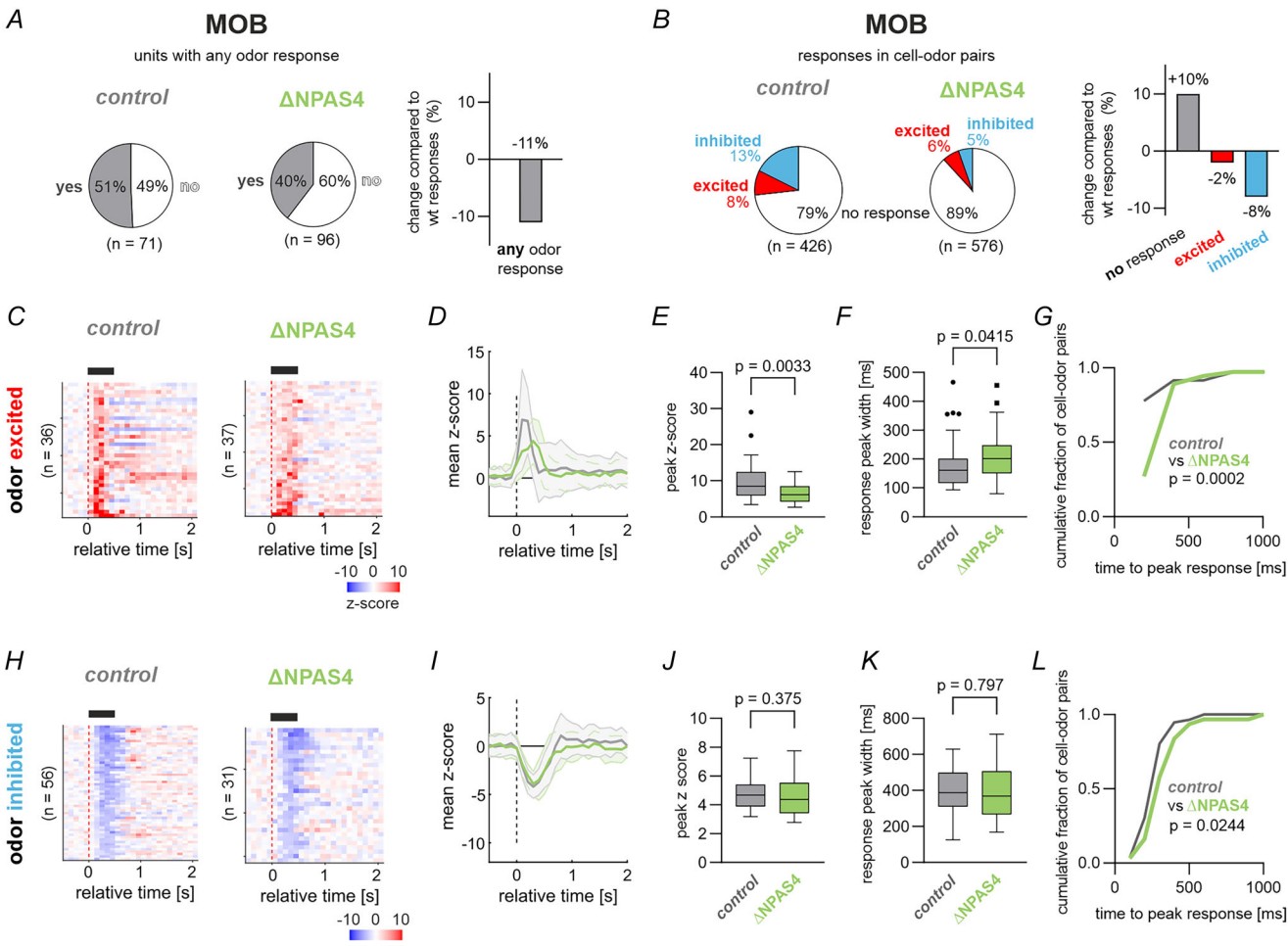

**Figure 3. $\Delta$NPAS4$^{M/T}$ impairs odour responses in the MOB.**
*A*, Pie charts indicate the fraction of units that showed a response (pooled excited or inhibited) to at least one of the six odors in the two genotypes (left) and their comparison (right). *B*, Pie charts indicate the fraction of odour-excited or odour-inhibited cell-odour pairs per genotype (left) and the comparison to control mice (right). A cell-odour pair was considered excited if the mean z-scored response in the odour window (0 to +0.5 s relative to odour onset) exceeded 1.96 (and analogously for odour-inhibited responses). *C–D*, Trial-averaged z-scored odour-excited responses (*C*; n = 36 unit-odour pairs for control and n = 37 unit-odour pairs for $\Delta$NPAS4$^{M/T}$) and their temporal evolution of the mean *z*-score (*D*; mean ± standard deviation). *E-G*, Comparison of features of the odour-excited responses between $\Delta$NPAS4$^{M/T}$ and control mice. The peak z-score (*E*), response peak width (*F*) and the time to peak response (*G*) were compared using two-sided Mann–Whitney *U*-tests (see results for test statistics). *H–I*, same as *C,D* for odour inhibited responses (n = 56 unit-odour pairs for control and n = 31 unit-odour pairs for $\Delta$NPAS4$^{M/T}$). *J–L*, Comparison of features of the odour-inhibited responses between $\Delta$NPAS4$^{M/T}$ and control mice. The peak *z*-score (*J*) and the time to peak response (*L*) were compared using two-sided Mann–Whitney *U*-tests. The response peak width (K) was compared using a two-sided unpaired *t* test. In this figure: Boxes show median and inter-quartile distance; whiskers extend to the most extreme points within 1.5 × IQR; points beyond are plotted individually.

In summary, selective developmental deletion of NPAS4 in M/Ts affected bulbar odour responses in the adult. M/Ts in mutants showed a pronounced loss of odour-inhibited responses and a reduced output strength in odour-driven firing with broadened responses. Specifically, M/Ts in mutant mice showed fewer odour responses and the relative number of inhibitory responses was decreased. Excitatory response displayed lower peak rates, and broader temporal dispersion. Overall, this leads to less specificity in the quality and temporal domain in mutant mice.

## Chemical similarity decoding is impaired in M/Ts of mutant mice

To quantify the effects of the mutation in adult mice, we analysed population responses in M/Ts. We therefore built concatenated population vectors (Laurent, 2002; Wolf, Hartig et al., 2024) across all recorded sessions and mice separately for each odour, respectively for control (Fig. 4*A1*) and mutant mice (Fig. 4*A2*). We considered all possible trial combinations and plotted the Euclidean distance of the population vector between two different aldehyde odours ($n = 19$ trials per odour). We hypothesized that the cross-odour distance would be larger than the trial-by-trial variability within the representation of the same odour (0 to +0.5 s relative to odour onset). We performed a statistical comparison of the distribution of distances between trials of a given odour (i.e. the trial-to-trial variability of the odour response) and the distribution of distances between the given and another odour which thereby informs on the discriminability of the two odour population responses (one-way Welch ANOVA with post-hoc correction for multiple comparisons using Games-Howell tests, adjusted *P*-values reported for every comparison, family-wise alpha threshold: $\alpha = 0.05$). In control mice, the representations of most aldehyde odors differentiated significantly from other aldehydes in the MOB ($F(20, 2079) = 248$, $P < 0.0001$, Welch's ANOVA, for post tests see Fig. 4*A3* and Supporting Information). Such discrimination was largely preserved in the MOB of $\Delta$NPAS4$^{M/T}$ mice ($F(20,2078) = 79.9$, $P < 0.0001$, Welch's ANOVA, for post tests see Fig. 4*A4*). Thus, M/Ts both in control and mutant mice generally discriminated different aldehydes.

A similar picture emerged when comparing the prediction accuracy of a classifier trained to decode the odour identity from the population activity. To compare between conditions with different unit sample sizes, we used subsampling from the total M/Ts sample separately for each genotype and repeatedly trained separate Gaussian Mixture Models or k-nearest neighbour classifiers (Fig. 4*B*; see methods for details). For statistical comparison between genotypes, we then used a generalized linear mixed effects model to predict the classifier accuracy from the mutant or control in the MOB. Overall, decoding accuracy is higher in control mice (Tukey's post-hoc test with Bonferroni correction for multiple comparisons, $\Delta$NPAS4$^{M/T}$ *vs.* control: $P < 0.0001$ for both Gaussian Mixture Models and k-nearest neighbour classifiers). Thus, MOB units in $\Delta$NPAS4$^{M/T}$ mice performed only slightly worse than in control mice.

We therefore tested next at the single-unit level, whether the difference in within-unit response difference to aldehydes is shallower in mutant than control mice (Fig. 4*C*). Specifically, we computed the distribution of response amplitudes to each odorant for each M/T and ranked them by their relative within-unit amplitudes. A steeper fall off may support more odour-specific responses and thereby better discrimination between odorants. The steepness of the response differences was shallower in mutants compared to controls ($F (1, 825) = 24.0$, $P < 0.0001$, 2-way repeated-measures ANOVA). Post-hoc test results were indicated as exact p-values (Šídák multiple comparison test, Fig. 4*C*).

We then tested to what extent M/Ts differentiate between aldehydes based on the difference on carbon chain length (Chaudhury et al., 2009). The goal was thus to test if the M/T population coded for chemical similarity between aldehydes. We therefore plotted unit response amplitudes as a function of the best response and difference in carbon chain length (Fig. 4*D*). This takes into account the fact that the best response is a characteristic of individual neurons. In a 2-way ANOVA, there was a significant main effect for chain length difference ($F (4, 825) = 17.27$, $P < 0.0001$) as well as a significant main effect of genotype ($F(1,825) = 31.03$, $P < 0.0001$). Šídák multiple comparison tests were indicated as adjusted p-values if significant (Fig. 4*D* and see Supporting Information). Post-hoc tests between odours were significant in controls, but not in mutant mice. As a consequence, we conclude that while the response amplitude of M/Ts in control mice is a function of different carbon chain length, those of mutant mice is not.

In summary, while M/Ts in both the control and mutant mice show discriminable odour responses, the encoding of chemical similarity by graded rate responses was lost as a consequence of the NPAS4 mutation.

## Odour coding in the AON of mutant mice

The site of gene deletion is in the MOB (cf. Fig. 1). We next wondered about the impact of these bulbar changes on downstream aldehyde coding in the olfactory cortex. We chose one particular target of bulbar output, the AON, to test how modulation of bulbar output affects

cortical coding (Fig. 5*A*). We implanted a set of mice with chronic tetrode arrays with 64 channels in the pars centralis of the AON (exemplary histology shown in Fig. 5*B* and summary in Fig. 5*C*). In the AON of mutant mice, we recorded a sample of 160 units. Single-units had comparable baseline firing rates (Fig. 5*D1*) and spike waveforms (Fig. 5*D2*) in the AON across genotypes ($n = 108$ units for control and $n = 160$ units for $\Delta$NPAS4$^{M/MT}$). A fraction of these units showed excited

or inhibited responses to each of the aldehydes (Fig. 5*E*). Z-scored responses of the five units with the highest average pairwise discriminability index (i.e. consistently different odour responses; for details see Method section '*Analysis of single-unit activity*') from control (Fig. 5*F*) and mutant mice (Fig. 5*G*) show relatively small differences between odours in the mutant (Fig. 5*G*). Note, that also high amplitude responses in mutants were observed, they differed from the control however in that they were much

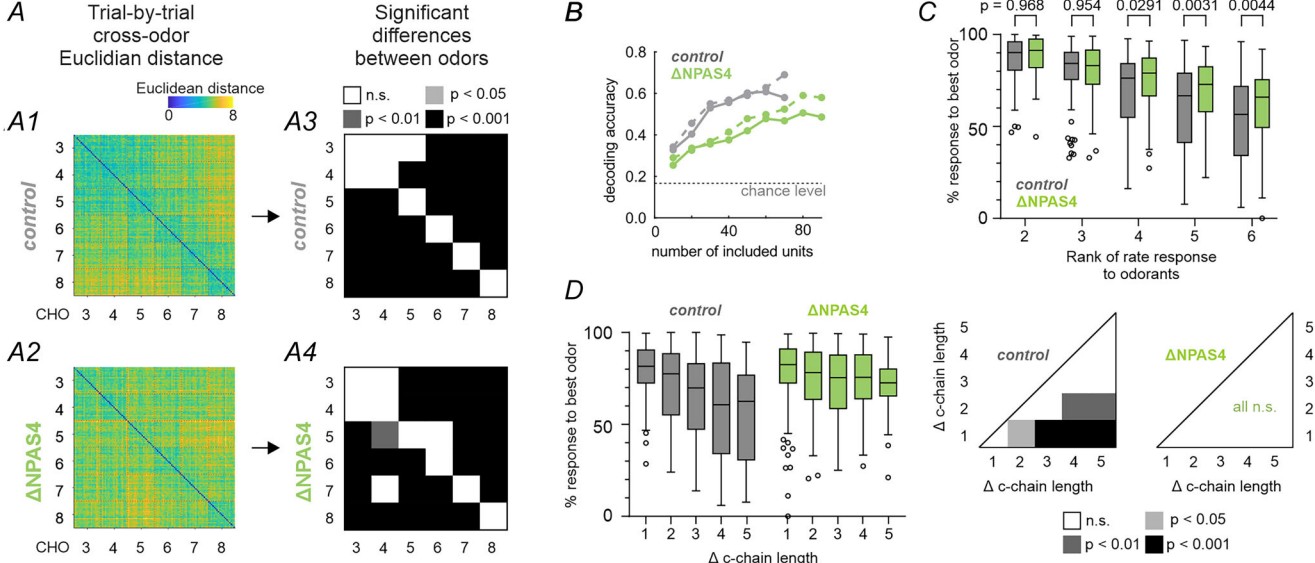

**Figure 4. Odour decoding in the MOB of $\Delta$NPAS4$^{M/T}$ mice.**
*A*, The Euclidean distance between spike count population vectors (0 to +0.5 s relative to odour onset) of different trials were computed for all combinations of trials of (A1) control and (A2) mutant mice ($n = 19$ trials per odour). We tested if neural populations differentiated between trials using a one-way Welch ANOVA. To test whether the cross-odour distance was larger than the trial-by-trial variability (i.e. distinctive encoding of the odours) in (A3) control and (A4) mutant mice, we performed post-hoc Games-Howell tests ($\alpha = 0.05$). The off-diagonal tiles indicate post-hoc test result of cross-odour distance against the trial-by-trial variability of the odour in that row. Exact *P*-values of post-hoc test results are provided in the Supporting Information file. *B*, Comparison of the odour decoding accuracy across genotypes. Separate Gaussian Mixture Models (solid lines) or k-nearest neighbor models (dashed lines) for every region and genotype were fit to the spike count population vector (response window from 0 to +0.5 s relative time). To allow for comparison of populations with different numbers of recorded populations, we used subsampling and repeated the analysis with increasing numbers of included units drawn from the full recorded population. For every number of included units, 500 random iterations of the training set were computed, and at each iteration the classifier accuracy was determined by testing the prediction on a trial which was omitted from the training set. The accuracies were compared between genotypes using generalized linear mixed effects models (see methods for details). Tukey's post-hoc test with Bonferroni correction for multiple comparisons confirms that in the MOB, decoding accuracy was significantly higher in the control than for $\Delta$NPAS4$^{M/T}$ mice ($\Delta$NPAS4$^{M/T}$ *vs.* control: $P < 0.0001$). *C*, The relative response strength per MOB unit (normalized to the strongest firing rate response) was averaged across the recorded populations and plotted as ranked response, to compare the range of response strength between mutants and control mice. Note the reduction in steepness of the response differences in mutants compared to controls in the MOB ($F_{(1, 825)} = 24.0$, $P < 0.0001$, 2-way repeated-measures ANOVA). Post-hoc tests were indicated as exact *P*-values (Šídák multiple comparison test). *D*, To test for chemical similarity coding, the relative response strength per MOB unit (normalized to the strongest firing rate response) was again averaged across the recorded populations. However, here it was plotted as a function of the difference in carbon chain length for mutants and control mice, respectively (left). In a 2-way ANOVA there was a significant main effect for chain length difference ($F_{(4,825)} = 17.27$, $P < 0.0001$) and there was a significant main effect of genotype ($F_{(1,825)} = 31.03$, $P < 0.0001$). Results of pairwise post-hoc Šídák multiple comparison tests were indicated in p-value matrix for controls and mutants (right; exact *P*-values of post-hoc test results are provided in the Supporting Information file). Post-hoc tests were significant in controls, but not in mutant mice. Thus, while chemical similarity coding was expressed in control MOB units, it was lost in mutants.

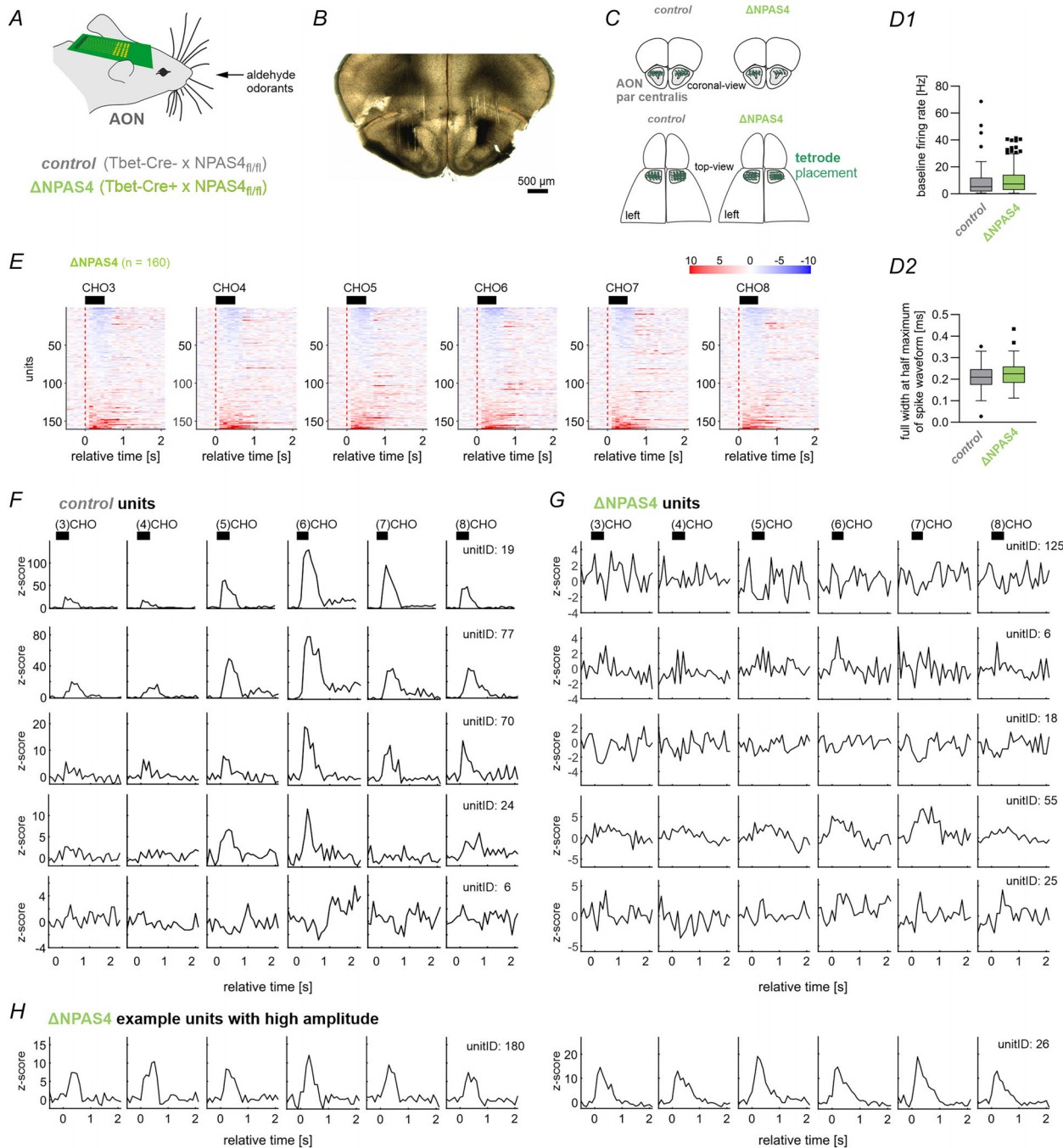

**Figure 5. Odour responses to aldehyde odorants in the AON.**
*A*, ΔNPAS4[M/T] or control mice with chronically implanted tetrode-arrays for single-unit recordings were placed in a head-fixed recording setup and presented aldehyde odorants with increasing lengths of the carbon chain (Propanal, (3)CHO, to Octanal, (8)CHO). One of the odors was presented for 500 ms in pseudorandomized order per trial. *B–C*, Histological section for confirmation of tetrode placement in the AON of an exemplary animal (B) and overview of tetrode locations in all control and mutant mice (C). *D*, The baseline firing rate (D1) and the median full-width at half maximum of spike waveforms (D2) of AON units of the two mutants or control mice ($n = 108$ units for control and $n = 160$ units for ΔNPAS4[M/T]). *E*, Trial-averaged *z*-scored responses of every unit from AON to the different odour types in ΔNPAS4[M/T] mice ($n = 160$ units). For each odour, the units were sorted based on their mean *z*-score during 0 to +0.5 s relative to odour onset. For matching response profiles in control mice see Wolf, Oettl et al. (2024). *F,G*, The *z*-scored responses to the different odours of the five responsive units with the highest average discriminability index d′ from control (F) and ΔNPAS4[M/T] (G). *H*, Exemplary *z*-scored responses from ΔNPAS4[M/T] with high-amplitude response, which show however more homogeneous responses across odorants compared to control animals (cf. F). In this figure: Boxes show median and inter-quartile distance; whiskers extend to the most extreme points within 1.5 × IQR; points beyond are plotted individually.

more homogeneous across odours (Fig. 5*H*, cf. 5F) To obtain quantitative genotype differences, we compared mutants directly to the control mice (units from control mice had been shown in Wolf, Oettl et al., 2024).

We first examined the unit responses to odorants (Fig. 6). ΔNPAS4$^{M/T}$ mice had a smaller proportion of units in the AON that responded to at least one of the aldehyde odorants (Fig. 6*A*). Specifically, the number of odour-excited cell-odour-pairs was reduced in mutants (Fig. 6*B*), consistent with the reduced output strength of excited M/Ts quantified in Fig. 3*D*-*G*. The remaining

odour-excited responses (Fig. 6*C*, $n = 184$ unit-odour pairs for control and $n = 146$ unit-odour pairs for ΔNPAS4$^{M/T}$) were on average smaller in rate amplitude in mutants than controls ($U = 11\,441$, $P = 0.0208$, MWU-Test, Fig. 6*D-E*). In these odour-excited responses, both the peak width (Fig. 5*F*) and latency to peak (Fig. 6*G*) displayed similar trends in mutants as in the MOB, however none of the them was significantly different compared to controls (peak width: $U = 11\,789$, $P = 0.0564$, MWU-Test; latency: $U = 12\,265$, $P = 0.167$, MWU-Test). Odour-inhibited units that were already sparse in the

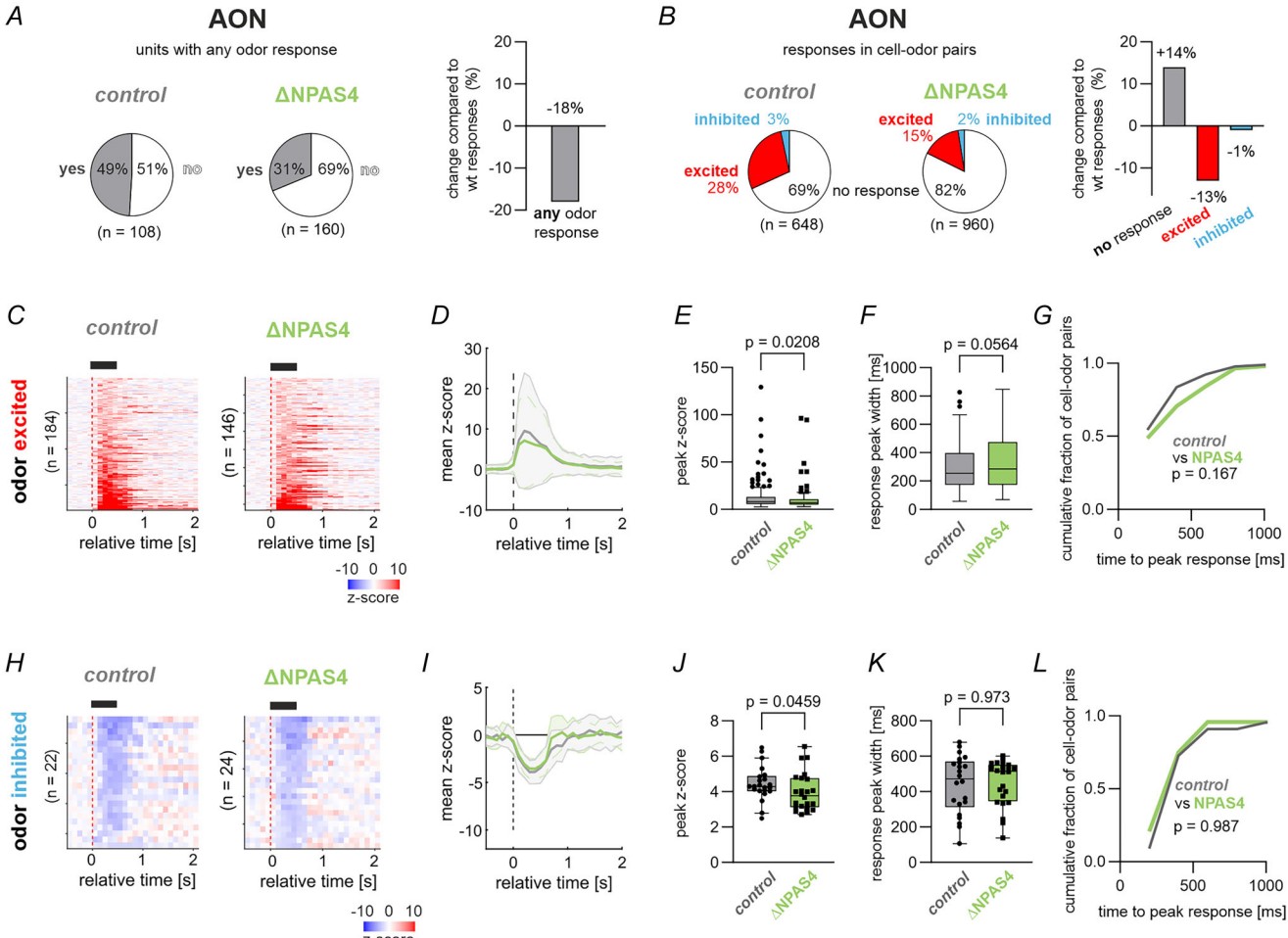

**Figure 6. ΔNPAS4$^{M/T}$ impairs odour responses in the AON.**
*A*, Pie charts indicate the fraction of AON units that showed a response (pooled excited or inhibited) to at least one of the six odours in the genotypes. *B*, Pie charts indicate the fraction of odour-excited or odour-inhibited cell-odour pairs in the genotypes. *C–D*, Trial-averaged z-scored odour-excited responses (*C*; $n = 184$ unit-odour pairs for control and $n = 146$ unit-odour pairs for ΔNPAS4$^{M/T}$) and their temporal evolution of the mean z-score (*D*; mean ± standard deviation). *E*, Comparison of features of the odour-excited responses between ΔNPAS4$^{M/T}$ and control mice. The peak *z*-score (left), response peak width (middle) and the time to peak response (right) were compared using two-sided Mann–Whitney *U*-tests (see results for test statistics). *F–G*, same as *C–D* for odour-inhibited responses ($n = 22$ unit-odour pairs for control and $n = 24$ unit-odour pairs for ΔNPAS4$^{M/T}$). *H*, Comparison of features of the odour-inhibited responses between ΔNPAS4$^{M/T}$ and control mice. The peak *z*-score (left), response peak width (middle) and the time to peak response (right) were compared using two-sided Mann–Whitney *U*-tests. In this figure: Boxes show median and inter-quartile distance; whiskers extend to the most extreme points within 1.5 × IQR; points beyond are plotted individually. Where $n < 30$, all data points are plotted individually.

AON of control mice (Fig. 6*H*, $n = 22$ unit-odour pairs for control and $n = 24$ unit-odour pairs for $\Delta NPAS4^{M/T}$), showed a slightly reduced peak response in mutants when compared to control mice (Fig. 6*I-J*; $U = 173$, $P = 0.0459$, MWU-Test), but were otherwise preserved in their average peak width (Fig. 5*K*; peak width: $U = 262$, $P = 0.973$, MWU-Test) and latency to peak (Fig. L; $U = 263$, $P = 0.987$, MWU-Test).

Using the recorded neurons as population data, we quantified the decoding of aldehyde odours in the AON. We focused on the discrimination of aldehydes from each other independent of chemical similarity as similarity coding is not a feature of the AON of wild-type mice (Wolf, Oettl et al., 2024). While all odours had significantly distinct representations from each other in the AON of control mice as evidenced by the trial-by-trial cross-odour Euclidean distances of the population responses (Fig. 7*A1*, $n = 19$ trials per odour), this discrimination was largely lost in the AON of $\Delta NPAS4^{M/T}$ mice where the distance between most odours did not exceed the intra-odour variability (Fig. 7*A2*). This was also observed with statistical testing. In control mice, most cross-odour comparisons differed significantly from intra-odour variability (F(20, 2076) = 101.7, $P < 0.0001$, Welch's ANOVA, for post tests see Fig. 7*A3* and Supporting Information). Such discrimination was largely lost in the AON of $\Delta NPAS4^{M/T}$ mice (F(20, 2082) = 9.7, $P < 0.0001$, Welch's ANOVA, for post tests see Fig. 7*A4*). Thus, M/Ts in mutant mice appeared to largely lose their ability to separate aldehydes in their population coding in the AON. We also revisited the coding of chemical similarity in the AON (Fig. 7*B*). As previously observed in an analysis of these recordings (Wolf, Oettl et al., 2024), we found no evidence for chemical similarity coding in the AON. As expected, the two-way ANOVA revealed a significant main effect of genotype (F(1,1330) = 90.35, $P < 0.0001$), which matches the overall reduction and uniformity of odour responses caused by NPAS4 deletion (cf. Fig. 7*A*). The main effect of carbon-chain length was however not significant (F(4,1330) = 1.443, $P = 0.2176$), and all Šídák-corrected post-hoc tests comparing responses across carbon-chain lengths were non-significant (see Fig. 7*B*, right), supporting that chemical similarity coding is not significantly expressed in the AON in either control or $\Delta NPAS4^{M/T}$ mice.

We therefore tested aldehyde classification in the coding of the population of single-units, again using Gaussian Mixture Model and k-nearest neighbours classifiers (Fig. 7*C*). In the AON of control mice, decoding accuracy of aldehyde odours improved with increasing unit numbers (Fig. 7*C*), consistent with a distributed odour coding of olfactory cortices. However, the decoding accuracy of AON units in $\Delta NPAS4^{M/T}$ mice remained at chance level independently of the sample size and

classifier model (one-sided Binomial test against chance level for highest number of included units, $P = 0.39$ (GMM), $P = 0.31$ (k-nearest neighbours), Fig. 7*C*). We therefore tested two potential contributions to this effect on the single-unit level. Firstly, one may expect that the intra-unit differences of response amplitudes are shallower. Relevant to this, single-units in the AON again showed graded response intensities to different aldehydes, while units recorded in the AON of mutants did not differentiate between aldehydes to the same degree (main effect of genotype: $F (1, 1330) = 155.4$, $P < 0.0001$, 2-way repeated-measures ANOVA). Post-hoc tests were indicated as exact p-values (Šídák multiple comparison test, Fig. 7*D*). Secondly, the distributions of maximum and mean pairwise discriminability indices (Fig. 7*E*) showed significantly smaller values in $\Delta NPAS4^{M/T}$ mice (two-sided Wilcoxon rank-sum test: $P < 0.001$ (mean d'), $P < 0.001$ (max d')). Taken together, the poor classification results in the pseudopopulation of all recorded units is likely a result of the combination of factors observed for the single-units in the mutant (lower fraction of excitatory responses, smaller amplitude of excitatory responses and more homogeneous, that is, less discriminable responses) leading to a decrease of signal-to-noise-ratio on the population level.

In summary, the MOB in control mice represented odours with rate code. Loss of NPAS4 selectively in M/Ts resulted in a bulbar network with impaired odour inhibited responses and loss of chemical similarity coding. As a consequence, in the downstream AON, the ability to discriminate between odours is impaired.

## Discussion

### Activity-dependent genetic factors in mitral cells determine odour response patterns

Here, we have investigated how an activity-dependent transcription factor determines the function of M/Ts in adulthood using conditional genetics to delete selectively the gene in these projection neurons but not in neighbouring cell-types. Compared to MOB interneurons, little is known about genetic factors that control the functional maturation of excitatory and inhibitory activity in M/Ts (Mizuguchi et al., 2012). The activity-induced transcription factor NPAS4 is expressed in M/Ts as well as in many other cell types both in the developing and adult brain (this study; Lin et al., 2008; Spiegel et al., 2014; Yoshihara et al., 2014). NPAS4 promotes the formation of inhibitory synapses onto cortical principal cells in an activity-dependent fashion, thereby controlling the circuit wiring (Lin et al., 2008; Spiegel et al., 2014). We showed that, in the adult, M/Ts with deleted NPAS4 predominantly lose odour-inhibited responses, with a slight reduction of odour-excited

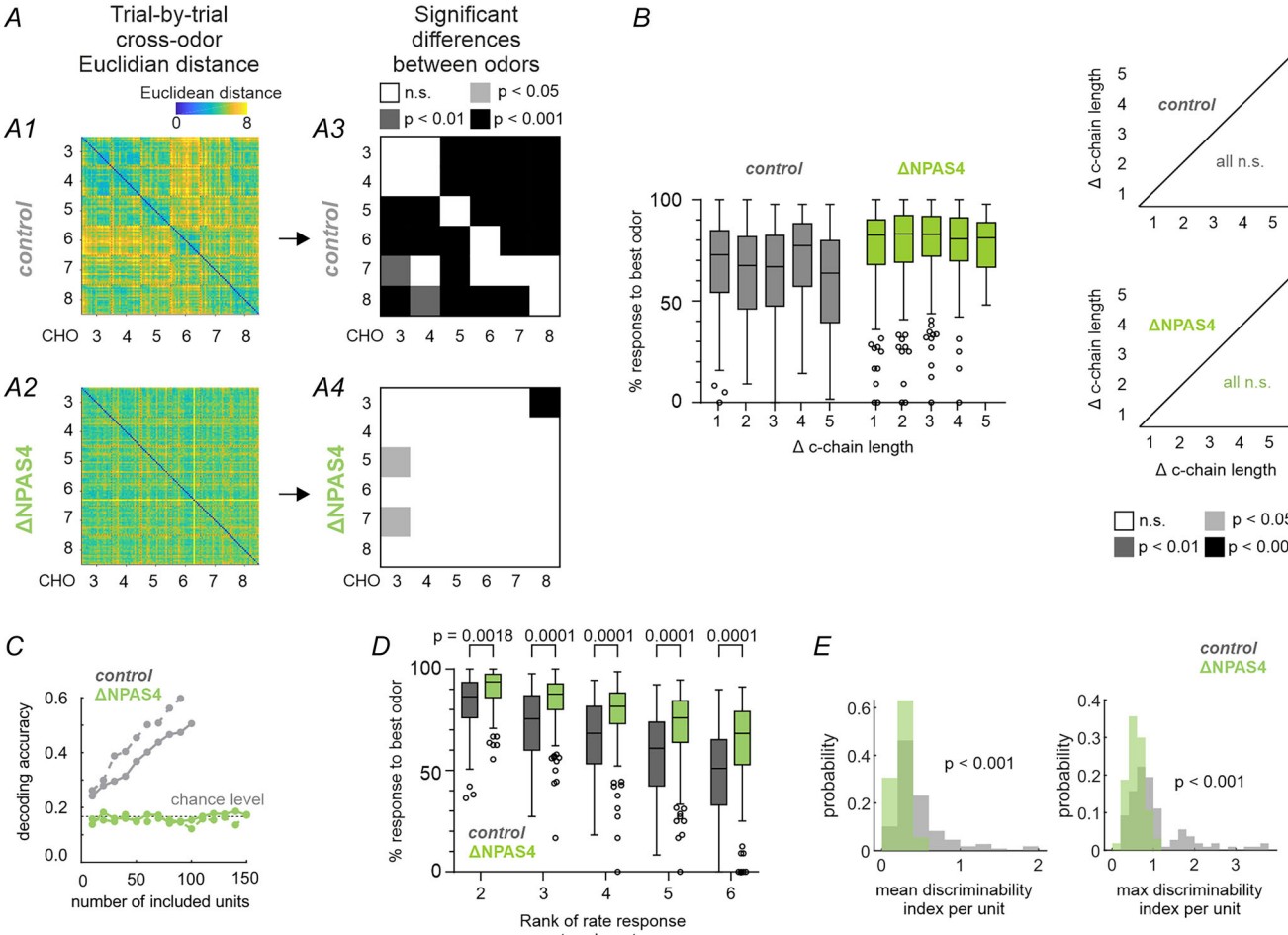

**Figure 7. Odour decoding in the AON of △NPAS4^M/T mice.**

*A*, The cross-odour Euclidean distance between spike count population vectors (0 to +0.5 s relative to odour onset) of different trials were computed for all combinations of trials in (A1) control and (A2) mutant mice. We tested if neural populations differentiated between trials using a one-way Welch ANOVA. To test whether the cross-odour distance was larger than the trial-by-trial variability (i.e. distinctive encoding of the odours) in (A3) control and (A4) mutant mice, we performed post-hoc Games-Howell tests ($\alpha = 0.05$). The off-diagonal tiles indicate post-hoc test result of cross-odour distance against the trial-by-trial variability of the odour in that row. Exact *P*-values of post-hoc test results are provided in the Supporting Information file. *B*, To test for chemical similarity coding, the relative response strength per AON unit (normalized to the strongest firing rate response) was averaged across the recorded populations and plotted as a function of the difference in carbon chain length for mutants and control mice, respectively (left; cf. Fig. 4D). In a 2-way ANOVA there was no significant main effect for chain length (see results for statistical tests). Results of pairwise post-hoc Šídák multiple comparison tests were indicated in p-value matrix for controls and mutants (right; exact p-values of post-hoc test results are provided in the Supporting Information file). None of the post-hoc tests were significant in either control or mutant mice, indicating no chemical similarity coding in the AON of control mice or in △NPAS4^M/T mice. *C*, Comparison of the odour decoding accuracy across genotypes in the AON with separate Gaussian Mixture Models (solid lines) and k-nearest neighbour models (dashed lines). In the AON, only in control mice, the classifier exceeded chance level and the accuracy was significantly higher than in △NPAS4^M/T mice (△NPAS4^M/T *vs.* control: $P < 0.0001$ for both GMM and k-nearest neighbour classifiers). *D*, Relative response strength per AON unit was plotted as ranked response to compare the range of response strength between mutants and control mice. Note the reduction in steepness of the response differences in mutants compared to controls in the AON ($F(1, 1330) = 155.4$, $P < 0.0001$, 2-way repeated-measures ANOVA). Post-hoc tests were indicated as exact p-values (Šídák multiple comparison test). *E*, Distributions of mean (left) and maximum (right) discriminability indices per unit are shown for △NPAS4^M/T and control mice. Two-sided Wilcoxon rank sum tests were used to compare the distributions, indicating lower discriminability values in the mutant mice. In this figure: Boxes show median and inter-quartile distance; whiskers extend to the most extreme points within $1.5 \times IQR$; points beyond are plotted individually.

responses. Consistent with such impaired inhibition, the temporal precision of odour-excited responses is decreased in NPAS4-deficient M/Ts with slower and broader peak responses. While the present study served to identify the cortico-bulbar effects of $\Delta$NPAS4$^{M/T}$ on odour coding in awake mice, future studies will have to clarify the neurodevelopmental alterations that this selective gene deletion imposes on the synaptic functions and wiring architecture locally, and also potentially with downstream cortices.

## Bulbar and cortical decoding of chemically similar odour molecules

At the network level, the bulb codes for chemical similarity in that more dissimilar molecules are also represented more differently in wild-type rodents (Johnson & Leon, 2000; Wachowiak & Cohen, 2001; Wolf, Oettl et al., 2024). Translated to cellular coding, M/Ts generate lateral inhibition from glomerular inputs targeted by olfactory sensory neurons with different affinities to aldehydes (Arevian et al., 2008; Imamura et al., 1992; Margrie et al., 2001; Mori et al., 1992; Sato et al., 1994; Yokoi et al., 1995). Consequently, while one aldehyde excites a mitral cell, an aldehyde with an additional C-atom may inhibit the same cell (Fig. 2*E* this study; Yokoi et al., 1995). This creates a representation of the molecular features of aldehydes from highly differentiated M/T population activity composed of odour-inhibited and -excited responses. In NPAS4-deficient M/Ts, the normal balance of odour-excited and inhibited responses is shifted with a predominant loss of inhibitory components and temporally blurred odour-excited responses (cf. Fig. 3). Consistent with the idea that chemical similarity coding is enabled by lateral inhibition, the ability to code for chemical similarity of aldehydes was lost in NPAS4-deficient M/Ts (cf. Fig. 4*D*). Future studies may examine the synaptic changes emerging from deletion of NPAS4 in olfactory bulb projection neurons.

M/Ts have patterns with relatively sparse odour-excited responses to different aldehydes, compared to the downstream cortical representations (Wolf, Oettl et al., 2024). While complex odour-inhibited and odour-excited patterns dominate in the MOB, the majority of AON responses are monotonic and odour-excited. Recordings were performed in the AON without identifying the specific neuron-type. The majority of neurons in the AON are glutamatergic projection neurons with a smaller fraction of inhibitory neurons than in other olfactory cortices (Brunjes et al., 2005). The pseudopopulation thus likely contains a majority of excitatory neurons, but also some rare types of interneurons (Brunjes et al., 2005). In control animals, many AON units are responsive to multiple aldehydes, differentiating them with graded rate response intensities.

Remarkably, in the AON of $\Delta$NPAS4$^{M/T}$ mice, odour decoding by the rate code is impaired even for relatively dissimilar aldehydes. One may have speculated that the ability to decode is more robust in the cortex as the distributed coding may recover information. Interestingly, there is no indication that the ability to decode recovers with increasing numbers of neurons in the population (c.f. Fig. 7*C*). The ability to decode was also not compensated by increased sparseness, namely the lower percentage of recruited cortical neurons to any odour in the mutant mice. This may suggest that downstream decoding relies on a certain structure in the upstream bulbar odour representation space to recreate this information in the AON.

Our data is in agreement with experiments using temporary modulation of excitation/inhibition balance in mice and rats. For example, indirect manipulations of bulbar inhibitory activity by transient alterations of cholinergic or noradrenergic tone, as well as a decrease in adult-born inhibitory neurons in the MOB change the contrast in M/T cell coding of straight chain aliphatic odorants (including the aldehydes used here) (Chaudhury et al., 2009; Moreno et al., 2009). Pharmacological interventions aiming to enhance inhibitory contrast in the MOB increased behavioural perceptual discrimination of closely related odorants rats or mice (Chaudhury et al., 2009). Similarly, enhancing the activity of granule cells that mediate lateral inhibition through activation of centrifugal inputs increases the discriminability of odours both in M/T and behaviourally (Alonso et al., 2012; Lepousez & Lledo, 2013; Oettl et al., 2016). In the present study, we had therefore focused on the odour coding consequences of altered inhibitory synaptic wiring. Future studies may explore this in other mutants, and also with regard to their behavioural odour discrimination abilities. We have not tested olfactory perception in our mutant mice, yet, previous work would predict a rather strong effect on perceptual discrimination. While it is perceivable that other cortical areas would be less affected through manipulations performed here and could compensate, most previous data point to a correlation between bulbar processing and perceptual discrimination (Abraham et al., 2010; Alonso et al., 2012; Chaudhury et al., 2009; Lepousez & Lledo, 2013; Moreno et al., 2009; Oettl et al., 2016). Also, a behavioural and modelling study suggested that some impairments of bulbar processing cannot be compensated for in cortical areas (Devore et al., 2014). While using our set of monomolecular aldehyde odorants has the advantage of being able to embed the results in the large body of previous studies conducted with the same odorants, it remains to be confirmed that the findings generalize to all classes of structurally similar odorants.

## Conclusion

The selective deletion in M/T reveals NPAS4 as a critical factor in these neurons to generate a MOB network with efficient lateral inhibition and for differential encoding of a class of chemically similar molecules. It also highlights the limited ability of the cortex to recover information from the MOB that was moderately impaired. NPAS4 is part of autism gene networks. Genetic alterations are systemic and will act at early bottom-up circuits as they do in higher cortical areas. Sensory processing deficits and compensatory alterations in sensory exploration predict the severity of autism spectrum disorders (Gliga et al., 2015; Kern et al., 2006). This gene adds to other genetic and neuromodulatory factors that alter olfactory processing elsewhere in the olfactory system and come with autism related phenotypes (Eltokhi et al., 2021; Hartig et al., 2021; Oettl et al., 2016; Resendez et al., 2020; Wolf, Hartig et al., 2024). The marked effects of autism candidate genes on sensory information processing will influence higher cognitions.

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

## Additional information

### Data availability statement

The electrophysiology data presented in this manuscript are available upon request from the Lead Contact.

### Competing interests

The authors declare no conflict of interest.

### Author contributions

D.W. designed and performed research, analysed data, and wrote the paper. L.L.O. designed and performed research and analysed data. L.S.G. designed and performed histology. C. Linster. analysed data and wrote the paper. C. Lois. designed research and contributed unpublished reagents. W.K. designed research, analysed data, and wrote the paper. All authors revised the manuscript. All authors approved the final version of the manuscript; agree to be accountable for all aspects of the work in ensuring that questions related to the accuracy or integrity of any part of the work are appropriately investigated and resolved; and all persons designated as authors qualify for authorship, and all those who qualify for authorship are listed.

### Funding

This work was supported by the BMBF-NSF CRCNS grant 'Oxystate' BMBF 01GQ1708 to W.K. and NSF 1 724 221 to C.L., Boehringer Ingelheim Foundation grant 'Complex Systems' to W.K.

### Acknowledgements

We thank Cathrin Löb for technical assistance. Dr. Namasivayam Ravi for contributions to the recordings and Dr. Eleonora Russo for discussions.

### Author's present address

David Wolf: Department of Epileptology, University of Bonn, Bonn, 53127 Bonn, Germany.

## Keywords

decoding, inhibition, olfactory bulb, olfactory cortex, transcription factor

## Supporting information

Additional supporting information can be found online in the Supporting Information section at the end of the HTML view of the article. Supporting information files available:

**Peer Review History**
**Supporting Information**

