## [Peer Review History · The Journal of Physiology]

Deletion of NPAS4 in olfactory bulb principal neurons alters E/I balance and impairs decoding of chemically similar odor molecules

Wolfgang Kelsch, David Wolf, Lars-Lennart Oetl, Luis Sanchez-Guardado, Christiane Linster, and Carlos Lois
DOI: 10.1113/JP288011

Corresponding author(s): Wolfgang Kelsch (wokelsch@uni-mainz.de)

Review Timeline:

Submission Date:	30-Oct-2024
Editorial Decision:	05-Dec-2024
Revision Received:	11-Aug-2025
Editorial Decision:	04-Sep-2025
Revision Received:	25-Sep-2025
Editorial Decision:	20-Oct-2025
Revision Received:	09-Nov-2025
Accepted:	13-Nov-2025

Senior Editor: Nathan Schoppa

Reviewing Editor: Nathan Schoppa

Transaction Report:

Dear Dr Kelsch,

Re: JP-RP-2024-288011 "**NPAS4 is required in mitral cells for the expression of lateral inhibition to decode chemically similar odor molecules**" by Wolfgang Kelsch, David Wolf, Lars-Lennart Oetl, Luis Sanchez-Guardado, Christiane Linster, and Carlos Lois

Thank you for submitting your manuscript to The Journal of Physiology. It has been assessed by a Reviewing Editor and by 2 expert referees and we are pleased to tell you that it is potentially acceptable for publication following satisfactory major revision.

LANGUAGE EDITING AND SUPPORT FOR PUBLICATION: If you would like help with English language editing, or other article preparation support, Wiley Editing Services offers expert help, including English Language Editing, as well as translation, manuscript formatting, and figure formatting at www.wileyauthors.com/eoo/preparation. You can also find resources for Preparing Your Article for general guidance about writing and preparing your manuscript at www.wileyauthors.com/eoo/prepresources.

REVISION CHECKLIST:

We look forward to receiving your revised submission.

Yours sincerely,

Nathan Schoppa
Senior Editor
The Journal of Physiology

REQUIRED ITEMS

- Author photo and profile. First or joint first authors are asked to provide a short biography (no more than 100 words for one author or 150 words in total for joint first authors) and a portrait photograph. These should be uploaded and clearly labelled together in a Word document with the revised version of the manuscript. See Information for Authors for further details.

- You must start the Methods section with a paragraph headed Ethical approval (https://jp.msubmit.net/cgi-bin/main.plex?form_type=display_requirements#methods).

Research must comply with The Journal's policies regarding animal experiments (<https://physoc.onlinelibrary.wiley.com/hub/animal-experiments>) and adherence to these policies must be stated in the manuscript.

Authors should confirm in their Methods section that their experiments were carried out according to the guidelines laid down by their institution's animal welfare committee, including an ethics approval reference number. The Methods section must contain a statement about access to food, water and housing, details of the anaesthetic regime: anaesthetic used, dose and route of administration, and method of killing the experimental animals.

- Your manuscript must include a complete Additional Information section, including competing interests; funding; author contributions and acknowledgements.

- The Journal of Physiology funds authors of provisionally accepted papers to use the premium BioRender site to create high resolution schematic figures. Follow this link and enter your details and the manuscript number to create and download figures. Upload these as the figure files for your revised submission. If you choose not to take up this offer, we require figures to be of similar quality and resolution. If you are opting out of this service to authors, state this in the Comments section on the Detailed Information page of the submission form. The link provided should only be used for the purposes of this submission. Authors will be charged for figures created on this premium BioRender account if they are not related to this

manuscript submission.

- Please upload separate high-quality figure files via the submission form.

- Please ensure that the Article File you upload is a Word file.

Reviewing Editor's comments:

Thank you for submitting your manuscript describing changes in odor-evoked responses in NPAS4 mutant mice for potential publication in Journal of Physiology. The manuscript has been reviewed by two expert reviewers, who felt that the work was addressing an interesting question and that many of the described changes in odor-evoked responses were generally convincing. Both reviewers however raised major concerns. All of the points that they raised should be addressed in a revised manuscript, which will need to be re-evaluated by both reviewers. The most important points included:

(1) The presentation of the Results needs to be substantially improved. As pointed out by Reviewer 2, the Results section is very short and does not fully describe the panels in the figures, making it difficult to follow the line of argumentation. Also, Reviewer 1 points out a number of ambiguities in the text, for example what is meant by "chemical similarity coding." In addition, I found some sentences very confusing, for example around lines 405-406: "The NPAS4 mutation selectively in M/Ts resulted in their bulbar network mainly in impaired odor inhibited responses and loss of chemical similarity coding." The authors should rewrite the Results section, fully describing the figures and generally making the manuscript easier to understand for non-experts in the field.

(2) Both reviewers pointed out that the study does not provide mechanistic information about how removing a transcription factor leads to altered odor-evoked responses, for example whether there are changes in inhibitory synapses and/or excitation/inhibition balance or presynaptic changes. While experiments that address these issues are not required, the scholarship of the work will be significantly improved if the authors discuss what they believe what might be going on mechanistically (supported by literature) and future experiments to test their ideas. In addition, the authors should speculate in the Discussion what they might predict for olfactory behavior based on their results (to address a concern from Reviewer 2).

(3) As outlined by Reviewer 2, some additional methodological information is needed, including how the authors verified positioning of the recording electrodes in OB and AON and how they controlled for oversampling in the longitudinal recordings. For the positioning of electrodes in OB, they indicate that they tested for "MT features" for the units, but what were these and can the authors reference a study that describes these?

(4) As pointed out by Reviewer 1, the loss of the ability of AON to decode structurally similar aldehydes in the mutant mice seems surprising given that differences in odor-evoked responses still appear to occur. Unless the authors have an explanation for this apparent discrepancy, they should test at least one other type of decoding model (other than Gaussian mixture) to see if it performs as poorly in their mutant mice.

(5) There should be more discussion of the limitations of the study, for example the fact that the authors only test aldehydes and no markers of cell-type are used in their AON recordings.

(6) Please check to make sure that terminal procedures for all of the test animals are described. I was not able to find them in the Methods section for the mice in which in vivo recordings were performed.

Referee #1:

In this paper, Wolf and colleagues examine the functional consequences of the loss of a transcription factor (NPAS4) in mitral and tufted (M/T) cells in the olfactory bulb of mice. NPAS4 has been studied previously in many other brain regions and shown to influence inhibitory synapse formation onto glutamatergic neurons. In this current work, the authors rationalized that NPAS4 may have similar effects on M/T cells. An elegant way of judging whether inhibition is altered is to examine odor-evoked responses, which have both increases and decreases in firing rates - should inhibition be reduced, there will be preferred effect on firing rate reductions. Indeed, the authors found that mitral cells respond to odors with less temporal precision, and with fewer inhibitory responses. This leads to more overlap in representation of different odors and poorer discrimination among closely related (by chemistry) odors.

Overall, this is a fine paper that adds significant new information about olfactory information processing. However, I find the large jump in the levels of investigation - from transcription factor to odor responses - a bit jarring. Of course, there are plausible connections at all the levels between - gene expression in mitral cells can change inhibitory synapse formation (is that dependent on the presynaptic activity as well?), which will change the overall circuit connectivity in some way, leading to functional changes. It feels like an important intermediate step of asking how loss of NPAS4 affects lateral and auto-inhibition using slice physiological or other methods might be valuable. That is, of course an entirely different study, but the authors might offer some more detailed logic of their hypotheses in the Discussion.

Specific Comments

The title refers to mitral cells, but the removal of NPAS4 is in all principal cells in the OB - I realize that the actual functional analysis is in the mitral cells, but they might consider telegraphing this more appropriately in the title and abstract.

Abstract: the transition between the second and third sentences seem very abrupt. Might consider being more logically smooth if you mention connectivity that leads to information processing, and then how transcription regulates connectivity.

Cross-odor distance: the characterization of similarity (or its complement, distance) is done using mitral and AON populations. It will be interesting to know how similar the odor distances are in the receptor neurons. In all analysis of the type done here using aldehydes, the assumption has been that odors with similar carbon chain length activate more similar patterns in the input layer - would be nice to acknowledge this, and cite any prior work that suggest this.

Line 365: "the few remaining inhibited responses...". This implies that only a small number of inhibited responses exist, but from Fig 3F we see that it is $n=31$ compared to 56, so not that small?

Line 388: "collapse of the classifier performance" - but the data shown so far, and text above indicate that the classifier performance is only slightly impaired in NPAS4 knockout?

Line 403 and elsewhere: It is not clear what the authors mean by "chemical similarity coding" - I think they mean that similar odors are DISTINCTLY coded (ie, discriminated better by neural activity)? Please make this clearer the reader.

AON data: the loss of decodability from AON activity in the mutant mouse seems striking, especially since it is only slightly impaired in the OB. I found this particularly puzzling since the data in Figure 5E and F seem to indicated enough diversity of responses across odors and neurons to allow for decoding. Perhaps the authors can comment on it, and maybe use other types of decoding than Gaussian mixture model.

Referee #2:

NPAS4 is an activity-dependent transcription factor that regulates neuronal E:I balance. The authors point out that the function of NPAS4 in early olfactory processing is unknown. They set about to investigate its role by selectively deleting NPAS4 in Tbet-Cre positive cells, a marker for a subpopulation of excitatory neurons in the olfactory bulb. The authors then performed electrophysiological recordings from early olfactory processing areas in awake mice and analyzed odour-evoked activity in mutant and control mice. According to the authors, chemical similarity tuning was affected at the level of the OB in mutant mice while also cortical odour representations decreased.

The general questions are interesting, and while this manuscript could have potential, many aspects seem lacking.

Major points:

- There is a large mismatch of text devoted to each section, e.g. the M/M part is as large or even larger as the super-short result section. Some figure legends comprise more text and are more informative than the figure's description in the main text (subfigures are sometimes not even mentioned in the results section). This makes it nearly impossible to follow the line of argumentation.
- The behavioural relevance of the findings is completely unclear
- Other parts of the olfactory cortex (e.g. piriform) might also be affected or, on the other hand, might compensate for the observed effects (so that the impact on behaviour would be minimal).
- Does the observed effect generalize the other odours?
- Tbx21 is not specific to MT cells and is also expressed in external tufted cells (e.g. Kosaka and Kosaka 2012).
- No histology for electrode position verification is provided
- The mechanisms of how NPAS4 would affect the excitatory inhibitory balance / presynaptic signalling is completely unclear from the MS
- How is oversampling the same neuronal population in longitudinal recordings controlled?
- No cell type-specific recordings; the cortical neuronal population is likely a mixed population of principal excitatory and local inhibitory neurons affecting or even explaining the observed variance

Minor points:

- The number of experiments should be reported more clearly throughout the MS.
- It is unclear why the authors expect that NPAS4 is expressed in a fraction of TBR2-positive cells.
- "The odor-inhibited responses (Fig. 3F-H) peaked later in mutants ($U = 630.5$, $p = 0.020$, MWU-Test)". Where can this be seen?
- Line 368 "Thus, in summary, selective developmental deletion of NPAS4 in M/Ts affected bulbar and cortical odor responses in the adult". Cortical responses have not been analysed at this part of the MS.
- Graded shading would be preferred compared to simple binary plots in e.g. Fig 4a /7a right.

END OF COMMENTS

Reviewing Editor's comments:

Thank you for submitting your manuscript describing changes in odor-evoked responses in NPAS4 mutant mice for potential publication in Journal of Physiology. The manuscript has been reviewed by two expert reviewers, who felt that the work was addressing an interesting question and that many of the described changes in odor-evoked responses were generally convincing. Both reviewers however raised major concerns. All of the points that they raised should be addressed in a revised manuscript, which will need to be re-evaluated by both reviewers. The most important points included:

(1) The presentation of the Results needs to be substantially improved. As pointed out by Reviewer 2, the Results section is very short and does not fully describe the panels in the figures, making it difficult to follow the line of argumentation. Also, Reviewer 1 points out a number of ambiguities in the text, for example what is meant by "chemical similarity coding." In addition, I found some sentences very confusing, for example around lines 405-406: "The NPAS4 mutation selectively in M/Ts resulted in their bulbar network mainly in impaired odor inhibited responses and loss of chemical similarity coding." The authors should rewrite the Results section, fully describing the figures and generally making the manuscript easier to understand for non-experts in the field.

We thank the reviewers and editor for the feedback and helpful comments. We addressed them accordingly. We thoroughly revised the results section, now fully describing the figures and improving the readability.

The line numbers indicated in this reply refer to the numbers of the attached manuscript .pdf with highlighted changes in red.

(2) Both reviewers pointed out that the study does not provide mechanistic information about how removing a transcription factor leads to altered odor-evoked responses, for example whether there are changes in inhibitory synapses and/or excitation/inhibition balance or presynaptic changes. While experiments that address these issues are not required, the scholarship of the work will be significantly improved if the authors discuss what they believe what might be going on mechanistically (supported by literature) and future experiments to test their ideas. In addition, the authors should speculate in the Discussion what they might predict for olfactory behavior based on their results (to address a concern from Reviewer 2).

We thank the editor for pointing this out. We revised the relevant section in the introduction, l. 102: "Here, we tested the consequences of a developmental gene deletion in M/Ts of a transcription factor that preferentially affects the maturation of inhibition in the neocortex. The activity-dependent transcription factor Neuronal PAS Domain Protein 4 (NPAS4) is required in cortical principal neurons to produce a balance between excitatory and inhibitory synapses (Fu et al., 2020). NPAS4 is responsive to excitation-coupled postsynaptic calcium influx (Bloodgood et al., 2013). **While the function of NPAS4 is not known for M/T maturation and their odor coding in the adult (Yoshihara et al., 2014; Ravi et al.,**

2017; Fujimoto et al., 2023), NPAS4 activates distinct programs of late-response genes in inhibitory and excitatory neocortical neurons (Lin et al., 2008; Spiegel et al., 2014). NPAS4 is rapidly induced in glutamatergic neurons in response to activity-dependent calcium influx (Lin et al., 2008; Bloodgood et al., 2013). Once expressed, NPAS4 regulates the transcription of specific downstream genes that promote the formation, maintenance, and function of inhibitory synapses onto these excitatory neurons (Lin et al., 2008). NPAS4 does not increase inhibition globally; it specifically enhances inhibitory input onto active excitatory neurons, allowing for precise homeostatic control for instance through compartment-specific BDNF expression (Bloodgood et al., 2013; Spiegel et al., 2014). We therefore probed this transcription factor to explore the role of inhibitory synaptic wiring on odor representations in M/T and the downstream AON in adult mice.”

We discuss potential behavioral consequences based on previous studies in the discussion, l. 620: “Our data is in agreement with experiments using temporary modulation of excitation/inhibition balance in mice and rats. For example, indirect manipulations of bulbar inhibitory activity by transient alterations of cholinergic or noradrenergic tone, as well as a decrease in adult-born inhibitory neurons in the MOB change the contrast in M/T cell coding of straight chain aliphatic odorants (including the aldehydes used here) (Chaudhury et al., 2009; Moreno et al., 2009). Pharmacological interventions aiming to enhance inhibitory contrast in the MOB increased behavioral perceptual discrimination of closely related odorants rats or mice (Chaudhury et al., 2009). Similarly, enhancing the activity of granule cells that mediate lateral inhibition through activation of centrifugal inputs increases the discriminability of odors both in M/T and behaviorally (Alonso et al., 2012; Lepousez and Lledo, 2013; Oettl et al., 2016). In the present study, we had therefore focused on the odor coding consequences of altered inhibitory synaptic wiring. Future studies may explore this in other mutants, and also with regard to their behavioral odor discrimination abilities. We have not tested olfactory perception in our mutant mice, yet, previous work would predict a rather strong effect on perceptual discrimination. While it is perceivable that other cortical areas would be less affected through manipulations performed here and could compensate, most previous data point to a correlation between bulbar processing and perceptual discrimination (Chaudhury et al., 2009; Moreno et al., 2009; Abraham et al., 2010; Alonso et al., 2012; Lepousez and Lledo, 2013; Oettl et al., 2016). Also, a behavioral and modeling study suggested that some impairments of bulbar processing cannot be compensated for in cortical areas (Devore et al., 2014).”.

(3) As outlined by Reviewer 2, some additional methodological information is needed, including how the authors verified positioning of the recording electrodes in OB and AON and how they controlled for oversampling in the longitudinal recordings. For the positioning of electrodes in OB, they indicate that they tested for "MT features" for the units, but what were these and can the authors reference a study that describes these?

In line with previous work (Kay & Laurent 1999; Rinberg et al. 2006) we only found detectable unit activity in the anatomical depth expected for the mitral cell layer. Recorded units showed similar firing rates (Fig. 2C1) as well as shape of spike waveforms (Fig. 2C2) as in previous studies (Kay and Laurent, 1999; Rinberg, 2006; Shusterman et al., 2011; Kollo et al., 2014).

We added in the methods section, l. 230: “Upon start of the recording period MOB drives were lowered until units with M/T features such as firing in rhythm with respiration, expected baseline firing rates (Fig.

2C1) and spike waveforms (Fig, 2C2) were detected in the anatomical depth expected for the mitral cell layer, consistent with previously described M/T features (Kay and Laurent, 1999; Rinberg, 2006; Shusterman et al., 2011; Kollo et al., 2014).”

And in the results section, l. 407: “Drives were lowered until units with M/T features such as firing in rhythm with respiration, expected baseline firing rates (Fig. 2C1) and spike waveforms (Fig, 2C2) were detected in the anatomical depth expected for the mitral cell layer (Kay and Laurent, 1999; Rinberg, 2006; Shusterman et al., 2011; Kollo et al., 2014).”

In the AON, the recording site was confirmed histologically as described below:

We added in l. 237: “*Histological confirmation of recording site*

For histological confirmation of recording sites in the AON after experiment completion, animals were anesthetized with ketamine/xylazine (300 mg/kg BW ketamine and 60 mg/kg BW xylazine diluted in 0,9% saline), followed by transcardial perfusion with 0.9% PBS and then 4% paraformaldehyde for fixation. Further immersion fixation in 4% paraformaldehyde of the whole skull for two weeks allowed visualization of the tetrode tracts. Serial coronal sections were prepared using a vibratome with 200 μ m slices for visualization of tetrode tracts. Images were acquired on a Axio Imager 2 (Zeiss).”

A typical image showing tetrode tracts in the AON after recordings with our chronic tetrode arrays, now in Figure 5B.

B

Revised caption to Fig. 5B: “**B-C**, Histological section for confirmation of tetrode placement in the AON of an exemplary animal (B) and overview of tetrode locations in all control and mutant mice (C).”

We ruled out that our results depended on oversampling in the longitudinal recordings. Putatively resampled units were identified by manual curation: units recorded on the same electrode, in the same animal in subsequent sessions that showed a similar waveform, odor response pattern and firing rate were excluded. In the wt sample 20/71 units were excluded for this control analysis, in the Δ NPAS4^{M/T} 15/96 units were excluded. The same results as in Fig. 3 were obtained without potential resampling (see new replotted Fig. 3 without putative resampling):

Same as Fig. 3 but upon removal of potential units that were resampled across multiple sessions in both genotypes. Specifically, in the present sample we removed units recorded on the same electrode, in the same animal in subsequent sessions that showed a similar waveform, odor response pattern and firing rate.

A, Pie charts indicate the fraction of units that showed a response (pooled excited or inhibited) to at least one of the six odors in the two genotypes (left) and their comparison (right).

B, Pie charts indicate the fraction of odor-excited or odor-inhibited cell-odor pairs per genotype (left) and the comparison to control mice (right). A cell-odor pair was considered excited if the mean z-scored response in the odor window (0 to +0.5 s relative to odor onset) exceeded 1.96 (and analogously for odor-inhibited responses).

C-D, Trial-averaged z-scored odor-excited responses (C) and their temporal evolution of the mean z-score (D).

E, Comparison of features of the odor-excited responses between Δ NPAS4M/T and control mice. The peak z-score (left), response peak width (middle) and the time to peak response (right) were compared using Mann-Whitney U-Tests (see results for test statistics).

F-G, same as C-D for odor inhibited responses.

H, Comparison of features of the odor-inhibited responses between Δ NPAS4M/T and control mice. The peak z-score (left) and the time to peak response (right) were compared using Mann-Whitney U-Tests. The response peak width (middle) was compared using an unpaired t-test.

In this figure: bars denote the median with whiskers extending to the 95% confidence interval.

We added in l. 290: “As the same animal was recorded more than once, we removed all M/T units that occurred on the same tetrode in subsequent sessions. Similar results were obtained replicating the same statistical differences as in Fig. 3 (data not shown), thereby ruling out the potential confounds originating from putative resampling.”

(4) As pointed out by Reviewer 1, the loss of the ability of AON to decode structurally similar aldehydes in the mutant mice seems surprising given that differences in odor-evoked responses still appear to occur. Unless the authors have an explanation for this apparent discrepancy, they should test at least one other type of decoding model (other than Gaussian mixture) to see if it performs as poorly in their mutant mice.

We thank the reviewer for highlighting this point. Upon re-evaluating the original visualizations (especially the response index for every cell-odor in the original ms. Fig. 5F), we noticed that the response index heatmap did not visualize the reasons for the impairment well. Specifically, the heatmap suggested a preserved graded response to different odors in the same cells. In response, we have replaced the previous Fig. 5F with a more targeted and interpretable comparison. Specifically, we now present side-by-side visualizations of the five odor-responsive units with the highest average pairwise discriminability index (d' , as defined below) from control animals (new Fig. 5F) and Δ NPAS4^{M/T} mutants (new Fig. 5G). We believe this revised presentation more clearly illustrates the differences in odor response selectivity between genotypes.

We revised the results, l. 501: “A fraction of these units showed excited or inhibited responses to each of the aldehydes (Fig. 5E). Z-score responses of the five units with the highest average pairwise discriminability index (i.e., consistently different odor responses; for details see Method section ‘Analysis of single-unit activity’) from control (Fig. 5F) and mutant mice (Fig. 5G) show relatively small differences between odors in the mutant (Fig. 5G). Note, that also high amplitude responses in mutants were observed, they differed from the control however in that they were much more homogeneous across odors (Fig. 5H, cf. 5F) To obtain quantitative genotype differences, we compared mutants directly to the control mice (units from control mice had been shown in Wolf et al., 2024b).”

We also confirmed that the result of decoding accuracy did not depend on the specific decoding model chosen. Repeating the analysis using multi-class error-correcting output codes (ECOC) models using binary k -nearest neighbors models (Matlab function ‘fitcecoc’) reproduces the same results obtained with Gaussian Mixture Models.

We added in l. 362: “To rule out that the observed loss of decoding accuracy in Δ NPAS4^{M/T} mutants in the AON depended on the specific decoding model chosen, we repeated the analysis as described above using multi-class error-correcting output codes (ECOC) models using binary k -nearest neighbors learners (Matlab function ‘fitcecoc’). This control analysis reproduced the same results obtained with Gaussian Mixture Models.”

We now show the results of both the GMM classifier and the k-nearest neighbor classifier in

New Fig. 4B: MOB

New Fig. 7B: AON

Fig. 4B (left, MOB) and Fig. 7B (right, AON). B, Comparison of the odor decoding accuracy across genotypes. Separate Gaussian Mixture Models (solid lines) or k-nearest neighbor models (dashed lines) for every region and mutant were fit to the spike count population vector (response window from 0 to +0.5 s relative time).

We also added another complementary analysis to characterize the loss of decoding accuracy in Δ NPAS4 units: For every unit, we computed all pairwise discriminability indices for the response during the odor presentation (0 to +0.5s relative time) and compared the distributions of average and maximum discriminability indices across units from Δ NPAS4 and control animals. We find significantly lower discriminability indices in the mutant mice (new Fig. 7D).

We added to the results I. 551: “Secondly, the distributions of maximum and mean pairwise discriminability indices (Fig. 7D) showed significantly smaller values in Δ NPAS4^{M/T} mice (two-sided Wilcoxon rank-sum test: $p < 0.001$ (mean d'), $p < 0.001$ (max d')). Taken together, the poor classification results in the pseudopopulation of all recorded units is likely a result of the combination of factors observed for the single-units in the mutant (lower fraction of excitatory responses, smaller amplitude of excitatory responses and more homogeneous, i.e. less discriminable responses) leading to a decrease of signal-to-noise-ratio on the population level.”

We added in the methods section: I. 316: “To quantify the discriminability of odor responses, we compared for every unit all discriminability indices for pairwise responses: $d' = \frac{|\mu_A - \mu_B|}{\sigma_{RMS}}$, where μ denotes the spike count during the stimulus presentation window (0 to +0.5 s relative to odor onset) across trials and σ_{RMS} denotes the root-mean-square value of the standard deviations of spike counts across trials. The resulting distributions of average and maximum discriminability indices per unit were compared between mutant and control mice using a two-sided Wilcoxon ranksum test.”

Thus, the poor classification results in the pseudopopulation of all recorded units is likely a results of the combination of factors observed for the single-units in the mutant (lower fraction of excitatory

responses, smaller amplitude of excitatory responses and more homogeneous, i.e. less discriminable responses) leading to a decrease of signal-to-noise-ratio on the population level (cf. Fig. 7A).

(5) There should be more discussion of the limitations of the study, for example the fact that the authors only test aldehydes and no markers of cell-type are used in their AON recordings.

We thank you for highlighting this. We added a section discussing the limitations of the current study and possible directions for future studies, l. 639.: “While using our set of monomolecular aldehyde odorants has the advantage of being able to embed the results in the large body of previous studies conducted with the same odorants, it remains to be confirmed that the findings generalize to all classes of structurally similar odorants.”

And, l. 577: “While the present study served to identify the cortico-bulbar effects of Δ NPAS4^{M/T} on odor coding in awake mice, future studies will have to clarify the neurodevelopmental alterations that this selective gene deletion imposes on the synaptic functions and wiring architecture locally, and also potentially with downstream cortices.”

And, l. 599: “Recordings were performed in the AON without identifying the specific neuron-type. The majority of neurons in the AON are glutamatergic projection neurons with a smaller fraction of inhibitory neurons than in other olfactory cortices (Brunjes et al., 2005). The pseudopopulation thus likely contains a majority of excitatory neurons, but also some rare types of interneurons (Brunjes et al., 2005).”

(6) Please check to make sure that terminal procedures for all of the test animals are described. I was not able to find them in the Methods section for the mice in which in vivo recordings were performed.

We thank you for pointing this out. We added in l. 169: “Mice were anesthetized with ketamine/xylazine (300 mg/kg BW ketamine and 60 mg/kg BW xylazine diluted in 0,9% saline).”

Referee #1:

In this paper, Wolf and colleagues examine the functional consequences of the loss of a transcription factor (NPAS4) in mitral and tufted (M/T) cells in the olfactory bulb of mice. NPAS4 has been studied previously in many other brain regions and shown to influence inhibitory synapse formation onto glutamatergic neurons. In this current work, the authors rationalized that NPAS4 may have similar effects on M/T cells. An elegant way of judging whether inhibition is altered is to examine odor-evoked responses, which have both increases and decreases in firing rates - should inhibition be reduced, there will be preferred effect on firing rate reductions. Indeed, the authors found that mitral cells respond to odors with less temporal precision, and with fewer inhibitory responses. This leads to more overlap in representation of different odors and poorer discrimination among closely related (by chemistry) odors.

Overall, this is a fine paper that adds significant new information about olfactory information processing. However, I find the large jump in the levels of investigation - from transcription factor to odor responses - a bit jarring. Of course, there are plausible connections at all the levels between - gene expression in mitral cells can change inhibitory synapse formation (is that dependent on the presynaptic activity as well?), which will change the overall circuit connectivity in some way, leading to functional changes. It feels like an important intermediate step of asking how loss of NPAS4 affects lateral and auto-inhibition using slice physiological or other methods might be valuable. That is, of course an entirely different study, but the authors might offer some more detailed logic of their hypotheses in the Discussion.

We thank the reviewer for their thoughtful and encouraging feedback. We agree that the transition from transcription factor function to changes in odor-evoked activity could benefit from a clearer articulation of the underlying rationale.

We chose to focus on NPAS4 because it is an immediate early gene known to play a central role in regulating the formation of inhibitory synapses onto excitatory neurons. Direct physiological measurements of lateral inhibition in NPAS4-deficient MOB circuits would be informative, such experiments fall outside the scope of this study as acknowledged by the reviewer. Odor representations in the MOB are strongly shaped by lateral inhibition among M/Ts, mediated by local inhibitory interneurons. These odor responses are then projected to downstream cortices. Here, we focus on how these odor responses are changed at the level of the output layer of the MOB and in a downstream olfactory cortex. In response, we have elaborated the manuscript to improve the logic of argumentation:

L. 577: **“While the present study served to identify the cortico-bulbar effects of Δ NPAS4^{M/T} on odor coding in awake mice, future studies will have to clarify the neurodevelopmental alterations that this selective gene deletion imposes on the synaptic functions and wiring architecture locally, and also potentially with downstream cortices.”**

And I. 620: “Our data is in agreement with experiments using temporary modulation of excitation/inhibition balance in mice and rats. For example, indirect manipulations of bulbar inhibitory activity by transient alterations of cholinergic or noradrenergic tone, as well as a decrease in adult-born inhibitory neurons in the MOB change the contrast in M/T cell coding of straight chain aliphatic odorants (including the aldehydes used here) (Chaudhury et al., 2009; Moreno et al., 2009). Pharmacological interventions aiming to enhance inhibitory contrast in the MOB increased behavioral perceptual discrimination of closely related odorants rats or mice (Chaudhury et al., 2009). Similarly, enhancing the activity of granule cells that mediate lateral inhibition through activation of centrifugal inputs increases the discriminability of odors both in M/T and behaviorally (Alonso et al., 2012; Lepousez and Lledo, 2013; Oetl et al., 2016). In the present study, we had therefore focused on the odor coding consequences of altered inhibitory synaptic wiring. Future studies may explore this in other mutants, and also with regard to their behavioral odor discrimination abilities. We have not tested olfactory perception in our mutant mice, yet, previous work would predict a rather strong effect on perceptual discrimination. While it is perceivable that other cortical areas would be less affected through manipulations performed here and could compensate, most previous data point to a correlation between bulbar processing and perceptual discrimination (Chaudhury et al., 2009; Moreno et al., 2009; Abraham et al., 2010; Alonso et al., 2012; Lepousez and Lledo, 2013; Oetl et al., 2016). Also, a behavioral and modeling study suggested that some impairments of bulbar processing cannot be compensated for in cortical areas (Devore et al., 2014).”

Specific Comments

The title refers to mitral cells, but the removal of NPAS4 is in all principal cells in the OB - I realize that the actual functional analysis is in the mitral cells, but they might consider telegraphing this more appropriately in the title and abstract.

We thank the reviewer for pointing this out and changed the title to: “NPAS4 is required in **olfactory bulb principal neurons** for the expression of lateral inhibition to decode chemically similar odor molecules”

Abstract: the transition between the second and third sentences seem very abrupt. Might consider being more logically smooth if you mention connectivity that leads to information processing, and then how transcription regulates connectivity.

We revised the beginning of the abstract:

“Odor representations are established in the olfactory bulb by a fine balance of excitatory and inhibitory activity. The projection neurons of the olfactory bulb, the mitral and tufted cells then pass this information to the olfactory cortices. While bulbar circuits have been studied at the neural and synaptic level, relatively little is known the about activity-dependent gene transcription machinery that shapes

connectivity of mitral/tufted cells and thereby discriminative bulbar odor representations. As a first step, we conditionally deleted a candidate gene involved in synaptic wiring selectively in mitral and tufted cells during embryonic development and performed single-cell recordings in the olfactory bulb and the anterior olfactory nucleus of adult awake mice.”

Cross-odor distance: the characterization of similarity (or its complement, distance) is done using mitral and AON populations. It will be interesting to know how similar the odor distances are in the receptor neurons. In all analysis of the type done here using aldehydes, the assumption has been that odors with similar carbon chain length activate more similar patterns in the input layer - would be nice to acknowledge this, and cite any prior work that suggest this.

We thank the reviewer for highlighting this point. We have incorporated this information into the introduction and appropriately acknowledged the prior work, l. 135:

“These aldehydes preferentially activate subsets of dorsomedial glomeruli (Imamura et al., 1992; Mori et al., 1992; Sato et al., 1994). **At the glomerular input level of the olfactory bulb, structurally related molecules elicit activation in overlapping sets of glomeruli** (Rubin and Katz, 1999; Johnson and Leon, 2000; Wachowiak and Cohen, 2001). **At the level of bulbar output**, certain aldehydes excite a M/T, while another closely related aldehyde inhibits the same M/T (Yokoi et al., 1995), an effect that emerges from lateral inhibition among M/T via dendro-dendritic inhibition of granule cell interneurons (Yokoi et al., 1995; Margrie et al., 2001; Arevian et al., 2008). **Perceptual discrimination between these odorants can be modulated via changes in excitation and inhibition** (Mandairon et al., 2006; Chaudhury et al., 2009; Abraham et al., 2010).”

Line 365: "the few remaining inhibited responses...". This implies that only a small number of inhibited responses exist, but from Fig 3F we see that it is n=31 compared to 56, so not that small?

We rewrote the results section and revised that statement, now l. 429: “**We then examined** the features of the odor-inhibited responses to these monomolecular odorants **in the two genotypes (Fig. 3H). The average response across M/Ts had a similar shape between genotypes (Fig. 3I). Specifically, when comparing the genotypes, the odor-inhibited responses** had similar amplitudes (U = 767, p = 0.375, MWU-Test, Fig 3J), similar peak width (t(85) = 0.3, p = 0.797, unpaired t-test, Fig. 3K), **but differed slightly in time to peak response (U = 630.5, p = 0.02, MWU-Test, Fig. 3L).”**

Line 388: "collapse of the classifier performance" - but the data shown so far, and text above indicate that the classifier performance is only slightly impaired in NPAS4 knockout?

We clarified that statement, l. 467: “**Overall, decoding accuracy is higher in control mice (Tukey’s post-hoc test with Bonferroni correction for multiple comparisons, Δ NPAS4^{M/T} vs control: p < 0.0001 for both Gaussian Mixture Models and k-nearest neighbor classifiers).**”

Line 403 and elsewhere: It is not clear what the authors mean by "chemical similarity coding" - I think

they mean that similar odors are **DISTINCTLY** coded (ie, discriminated better by neural activity)? Please make this clearer the reader.

We thank the reviewer for this clarification and now better explain this in the results section, l. 479: “We then tested to what extent M/Ts differentiate between aldehydes based on the difference on carbon chain length (Chaudhury et al., 2009). The goal was thus to test if the M/T population coded for chemical similarity between aldehydes. We therefore plotted unit response amplitudes as a function of the best response and difference in carbon chain length (Fig. 4D). This takes into account the fact that the best response is a characteristic of individual neurons. In a 2-way repeated-measures ANOVA, there was a significant main effect for chain length difference ($F(4, 825) = 17.27, p < 0.0001$) as well as a significant main effect of genotype ($F(1, 825) = 31.03, p < 0.0001$). Šídák multiple comparison tests were indicated as adjusted p-values if significant (Fig. 4D). Post-hoc tests between odors were significant in controls, but not in mutant mice. As a consequence, we conclude that while the response amplitude of M/Ts in control mice is a function of different carbon chain length, those of mutant mice is not. In summary, while M/Ts in both the control and mutant mice show discriminable odor responses, the encoding of chemical similarity by graded rate responses was lost as a consequence of the NPAS4 mutation.”

AON data: the loss of decodability from AON activity in the mutant mouse seems striking, especially since it is only slightly impaired in the OB. I found this particularly puzzling since the data in Figure 5E and F seem to indicated enough diversity of responses across odors and neurons to allow for decoding. Perhaps the authors can comment on it, and maybe use other types of decoding than Gaussian mixture model.

We thank the reviewer for highlighting this point. Upon re-evaluating the original visualizations (especially the response index for every cell-odor in the original ms. Fig. 5F), we noticed that the response index heatmap did not visualize the reasons for the impairment well. Specifically, the heatmap suggested a preserved graded response to different odors in the same cells. In response, we have replaced the previous Fig. 5F with a more targeted and interpretable comparison. Specifically, we now present side-by-side visualizations of the five odor-responsive units with the highest average pairwise discriminability index (d' , as defined below) from control animals (new Fig. 5F) and Δ NPAS4^{M/T} mutants (new Fig. 5G). We believe this revised presentation more clearly illustrates the differences in odor response selectivity between genotypes.

We revised the results, l. 501: “A fraction of these units showed excited or inhibited responses to each of the aldehydes (Fig. 5E). Z-score responses of the five units with the highest average pairwise discriminability index (i.e., consistently different odor responses; for details see Method section ‘Analysis of single-unit activity’) from control (Fig. 5F) and mutant mice (Fig. 5G) show relatively small differences between odors in the mutant (Fig. 5G). Note, that also high amplitude responses in mutants were observed, they differed from the control however in that they were much more homogeneous across odors (Fig. 5H, cf. 5F) To obtain quantitative genotype differences, we compared mutants directly to the control mice (units from control mice had been shown in Wolf et al., 2024b).”

We also confirmed that the result of decoding accuracy did not depend on the specific decoding model chosen. Repeating the analysis using multi-class error-correcting output codes (ECOC) models using binary k -nearest neighbors models (Matlab function 'fitcecoc') reproduces the same results obtained with Gaussian Mixture Models.

We added in l. 362: “To rule out that the observed loss of decoding accuracy in Δ NPAS4^{M/T} mutants in the AON depended on the specific decoding model chosen, we repeated the analysis as described above using multi-class error-correcting output codes (ECOC) models using binary k -nearest neighbors learners (Matlab function 'fitcecoc'). This control analysis reproduced the same results obtained with Gaussian Mixture Models.”

We now show the results of both the GMM classifier and the k -nearest neighbor classifier in

New Fig. 4B: MOB

New Fig. 7B: AON

Fig. 4B (left, MOB) and Fig. 7B (right, AON). B, Comparison of the odor decoding accuracy across genotypes. Separate Gaussian Mixture Models (solid lines) or k -nearest neighbor models (dashed lines) for every region and mutant were fit to the spike count population vector (response window from 0 to +0.5 s relative time).

We also added another complementary analysis to characterize the loss of decoding accuracy in Δ NPAS4 units: For every unit, we computed all pairwise discriminability indices for the response during the odor presentation (0 to +0.5s relative time) and compared the distributions of average and maximum discriminability indices across units from Δ NPAS4 and control animals. We find significantly lower discriminability indices in the mutant mice (new Fig. 7D).

We added to the results l. 551: “Secondly, the distributions of maximum and mean pairwise discriminability indices (Fig. 7D) showed significantly smaller values in Δ NPAS4^{M/T} mice (two-sided Wilcoxon rank-sum test: $p < 0.001$ (mean d'), $p < 0.001$ (max d')). Taken together, the poor classification results in the pseudopopulation of all recorded units is likely a result of the combination of factors observed for the single-units in the mutant (lower fraction of excitatory responses, smaller amplitude of

excitatory responses and more homogeneous, i.e. less discriminable responses) leading to a decrease of signal-to-noise-ratio on the population level.”

We added in the methods section: l. 316: “To quantify the discriminability of odor responses, we compared for every unit all discriminability indices for pairwise responses: $d' = \frac{|\mu_A - \mu_B|}{\sigma_{RMS}}$, where μ denotes the spike count during the stimulus presentation window (0 to +0.5 s relative to odor onset) across trials and σ_{RMS} denotes the root-mean-square value of the standard deviations of spike counts across trials. The resulting distributions of average and maximum discriminability indices per unit were compared between mutant and control mice using a two-sided Wilcoxon ranksum test.”

Thus, the poor classification results in the pseudopopulation of all recorded units is likely a results of the combination of factors observed for the single-units in the mutant (lower fraction of excitatory responses, smaller amplitude of excitatory responses and more homogeneous, i.e. less discriminable responses) leading to a decrease of signal-to-noise-ratio on the population level (cf. Fig. 7A).

Referee #2:

NPAS4 is an activity-dependent transcription factor that regulates neuronal E:I balance. The authors point out that the function of NPAS4 in early olfactory processing is unknown. They set about to investigate its role by selectively deleting NPAS4 in Tbet-Cre positive cells, a marker for a subpopulation of excitatory neurons in the olfactory bulb. The authors then performed electrophysiological recordings from early olfactory processing areas in awake mice and analyzed odour-evoked activity in mutant and control mice. According to the authors, chemical similarity tuning was affected at the level of the OB in mutant mice while also cortical odour representations decreased.

The general questions are interesting, and while this manuscript could have potential, many aspects seem lacking.

Major points:

- There is a large mismatch of text devoted to each section, e.g. the M/M part is as large or even larger as the super-short result section. Some figure legends comprise more text and are more informative than the figure's description in the main text (subfigures are sometimes not even mentioned in the results section). This makes it nearly impossible to follow the line of argumentation.

We thank the reviewer for the feedback and helpful comments. We thoroughly revised the results section, now fully describing the figures and improving the readability.

- The behavioural relevance of the findings is completely unclear

We added a paragraph on expected behavioural consequences based on previous studies in the discussion, l. 620: **“Our data is in agreement with experiments using temporary modulation of excitation/inhibition balance in mice and rats. For example, indirect manipulations of bulbar inhibitory activity by transient alterations of cholinergic or noradrenergic tone, as well as a decrease in adult-born inhibitory neurons in the MOB change the contrast in M/T cell coding of straight chain aliphatic odorants (including the aldehydes used here) (Chaudhury et al., 2009; Moreno et al., 2009). Pharmacological interventions aiming to enhance inhibitory contrast in the MOB increased behavioral perceptual discrimination of closely related odorants rats or mice (Chaudhury et al., 2009). Similarly, enhancing the activity of granule cells that mediate lateral inhibition through activation of centrifugal inputs increases the discriminability of odors both in M/T and behaviorally (Alonso et al., 2012; Lepousez and Lledo, 2013; Oettl et al., 2016). In the present study, we had therefore focused on the odor coding consequences of altered inhibitory synaptic wiring. Future studies may explore this in other mutants, and also with regard to their behavioral odor discrimination abilities. We have not tested olfactory perception in our mutant mice, yet, previous work would predict a rather strong effect on perceptual discrimination. While it is perceivable that other cortical areas would be less affected through**

manipulations performed here and could compensate, most previous data point to a correlation between bulbar processing and perceptual discrimination (Chaudhury et al., 2009; Moreno et al., 2009; Abraham et al., 2010; Alonso et al., 2012; Lepousez and Lledo, 2013; Oettl et al., 2016). Also, a behavioral and modeling study suggested that some impairments of bulbar processing cannot be compensated for in cortical areas (Devore et al., 2014).”

- Other parts of the olfactory cortex (e.g. piriform) might also be affected or, on the other hand, might compensate for the observed effects (so that the impact on behaviour would be minimal).

We addressed this point in the discussion, l. 632: “We have not tested olfactory perception in our mutant mice, yet, previous work would predict a rather strong effect on perceptual discrimination. While it is perceivable that other cortical areas would be less affected through manipulations performed here and could compensate, most previous data point to a correlation between bulbar processing and perceptual discrimination (Chaudhury et al., 2009; Moreno et al., 2009; Abraham et al., 2010; Alonso et al., 2012; Lepousez and Lledo, 2013; Oettl et al., 2016). Also, a behavioral and modeling study suggested that some impairments of bulbar processing cannot be compensated for in cortical areas (Devore et al., 2014).”

- Does the observed effect generalize the other odours?

We addressed this point in discussion, l. 639: “While using our set of monomolecular aldehyde odorants has the advantage of being able to embed the results in the large body of previous studies conducted with the same odorants, it remains to be confirmed that the findings generalize to all classes of structurally similar odorants.”

- Tbx21 is not specific to MT cells and is also expressed in external tufted cells (e.g. Kosaka and Kosaka 2012).

We now include this information, l. 394.: “At higher magnification, tdTomato expression was also observed in some putative external tufted cells (Fig. 1E2, 1E3), consistent with prior reports (Kosaka and Kosaka, 2012).”

- No histology for electrode position verification is provided

We added information on histological verification of tetrode position and now include an exemplary image of the AON in Fig. 5B. In the MOB, in line with previous work (Kay & Lauren 1999; Rinberg et al. 2006) we only found detectable unit activity in the anatomical depth expected for the mitral cell layer. Recorded units showed similar firing rates, shape and homogeneity of spike waveforms as in previous studies (Kay and Laurent, 1999; Rinberg, 2006; Shusterman et al., 2011; Kollo et al., 2014).

In the AON, the recording site was confirmed histologically as described below:

We added in l. 237: “*Histological confirmation of recording site*

For histological confirmation of recording sites in the AON after experiment completion, animals were anesthetized with ketamine/xylazine (300 mg/kg BW ketamine and 60 mg/kg BW xylazine diluted in 0.9% saline), followed by transcardial perfusion with 0.9% PBS and then 4% paraformaldehyde for fixation. Further immersion fixation in 4% paraformaldehyde of the whole skull for two weeks allowed visualization of the tetrode tracts. Serial coronal sections were prepared using a vibratome with 200 μ m slices for visualization of tetrode tracts. Images were acquired on a Axio Imager 2 (Zeiss).”

A typical image showing tetrode tracts in the AON after recordings with our chronic tetrode arrays, now in Figure 5B.

B

Revised caption to Fig. 5B: “**B-C**, Histological section for confirmation of tetrode placement in the AON of an exemplary animal (B) and overview of tetrode locations in all control and mutant mice (C).”

- The mechanisms of how NPAS4 would affect the excitatory inhibitory balance / presynaptic signalling is completely unclear from the MS

We now detailed this point in the introduction, l. 108: “The activity-dependent transcription factor Neuronal PAS Domain Protein 4 (NPAS4) is required in cortical principal neurons to produce a balance between excitatory and inhibitory synapses (Fu et al., 2020). NPAS4 is responsive to excitation-coupled postsynaptic calcium influx (Bloodgood et al., 2013). While the function of NPAS4 is not known for M/T maturation and their odor coding in the adult (Yoshihara et al., 2014; Ravi et al., 2017; Fujimoto et al., 2023), NPAS4 activates distinct programs of late-response genes in inhibitory and excitatory neocortical neurons (Lin et al., 2008; Spiegel et al., 2014). NPAS4 is rapidly induced in glutamatergic neurons in response to activity-dependent calcium influx (Lin et al., 2008; Bloodgood et al., 2013). Once expressed, NPAS4 regulates the transcription of specific downstream genes that promote the formation, maintenance, and function of inhibitory synapses onto these excitatory neurons (Lin et al., 2008). NPAS4 does not increase inhibition globally; it specifically enhances inhibitory input onto active excitatory neurons, allowing for precise homeostatic control for instance through compartment-specific BDNF expression (Bloodgood et al., 2013; Spiegel et al., 2014). We therefore probed this transcription factor to

explore the role of inhibitory synaptic wiring on odor representations in M/T and the downstream AON in adult mice.”

And in the discussion, l. 620: “Our data is in agreement with experiments using temporary modulation of excitation/inhibition balance in mice and rats. For example, indirect manipulations of bulbar inhibitory activity by transient alterations of cholinergic or noradrenergic tone, as well as a decrease in adult-born inhibitory neurons in the MOB change the contrast in M/T cell coding of straight chain aliphatic odorants (including the aldehydes used here) (Chaudhury et al., 2009; Moreno et al., 2009). Pharmacological interventions aiming to enhance inhibitory contrast in the MOB increased behavioral perceptual discrimination of closely related odorants rats or mice (Chaudhury et al., 2009). Similarly, enhancing the activity of granule cells that mediate lateral inhibition through activation of centrifugal inputs increases the discriminability of odors both in M/T and behaviorally (Alonso et al., 2012; Lepousez and Lledo, 2013; Oetl et al., 2016). In the present study, we had therefore focused on the odor coding consequences of altered inhibitory synaptic wiring. Future studies may explore this in other mutants, and also with regard to their behavioral odor discrimination abilities. We have not tested olfactory perception in our mutant mice, yet, previous work would predict a rather strong effect on perceptual discrimination. While it is perceivable that other cortical areas would be less affected through manipulations performed here and could compensate, most previous data point to a correlation between bulbar processing and perceptual discrimination (Chaudhury et al., 2009; Moreno et al., 2009; Abraham et al., 2010; Alonso et al., 2012; Lepousez and Lledo, 2013; Oetl et al., 2016). Also, a behavioral and modeling study suggested that some impairments of bulbar processing cannot be compensated for in cortical areas (Devore et al., 2014).”

- How is oversampling the same neuronal population in longitudinal recordings controlled?

We ruled out that our results depended on oversampling in the longitudinal recordings. Putatively resampled units were identified by manual curation: units recorded on the same electrode, in the same animal in subsequent sessions that showed a similar waveform, odor response pattern and firing rate were excluded. In the wt sample 20/71 units were excluded for this control analysis, in the Δ NPAS4^{M/T} 15/96 units were excluded. The same results as in Fig. 3 were obtained without potential resampling (see new replotted Fig. 3 without putative resampling):

Same as Fig. 3 but upon removal of potential units that were resampled across multiple sessions in both genotypes. Specifically, in the present sample we removed units recorded on the same electrode, in the same animal in subsequent sessions that showed a similar waveform, odor response pattern and firing rate.

A, Pie charts indicate the fraction of units that showed a response (pooled excited or inhibited) to at least one of the six odors in the two genotypes (left) and their comparison (right).

B, Pie charts indicate the fraction of odor-excited or odor-inhibited cell-odor pairs per genotype (left) and the comparison to control mice (right). A cell-odor pair was considered excited if the mean z-scored response in the odor window (0 to +0.5 s relative to odor onset) exceeded 1.96 (and analogously for odor-inhibited responses).

C-D, Trial-averaged z-scored odor-excited responses (C) and their temporal evolution of the mean z-score (D).

E, Comparison of features of the odor-excited responses between Δ NPAS4M/T and control mice. The peak z-score (left), response peak width (middle) and the time to peak response (right) were compared using Mann-Whitney U-Tests (see results for test statistics).

F-G, same as C-D for odor inhibited responses.

H, Comparison of features of the odor-inhibited responses between Δ NPAS4M/T and control mice. The peak z-score (left) and the time to peak response (right) were compared using Mann-Whitney U-Tests. The response peak width (middle) was compared using an unpaired t-test.

In this figure: bars denote the median with whiskers extending to the 95% confidence interval.

We added in l. 290: “As the same animal was recorded more than once, we removed all M/T units that occurred on the same tetrode in subsequent sessions. Similar results were obtained replicating the same statistical differences as in Fig. 3 (data not shown), thereby ruling out the potential confounds originating from putative resampling.”

- No cell type-specific recordings; the cortical neuronal population is likely a mixed population of principal excitatory and local inhibitory neurons affecting or even explaining the observed variance

We addressed this point in discussion, l. 599: “Recordings were performed in the AON without identifying the specific neuron-type. The majority of neurons in the AON are glutamatergic projection neurons with a smaller fraction of inhibitory neurons than in other olfactory cortices (Brunjes et al., 2005). The pseudopopulation thus likely contains a majority of excitatory neurons, but also some rare types of interneurons (Brunjes et al., 2005).”

Minor points:

- The number of experiments should be reported more clearly throughout the MS.

We added the number of recording sessions and mice per experiment, l. 263:

“A total of 167 units in the MOB (n = 71 units from 18 sessions in 5 control mice and n = 96 units from 18 sessions in 5 Δ NPAS4^{M/T} mice) and a total of 268 units in the AON (n = 108 units from 16 sessions in 9 control mice and n = 160 units from 11 sessions in 6 Δ NPAS4^{M/T} mice) were included in the analysis.”

- It is unclear why the authors expect that NPAS4 is expressed in a fraction of TBR2-positive cells.

NPAS4 is an immediate early gene, whose transcription is activity-dependent. Thus, under baseline conditions without odor stimulation we expected to find only a fraction of putative M/Ts to express NPAS4 at the timepoint of sacrificing the animal.

We clarified in the results, l. 385: “As expected for a gene with activity-dependent induction, NPAS4 protein expresses under baseline conditions in a fraction of Tbr2-immunoreactive cells in the mitral cell layer and also in the external plexiform and periglomerular layer at a snapshot at postnatal day 0 or in adult mice (Fig. 1A-C).”

- "The odor-inhibited responses (Fig. 3F-H) peaked later in mutants (U = 630.5, p = 0.020, MWU-Test)". Where can this be seen?

In the previous Fig. 3H (right), now Fig. 3L, the distribution of time to peak responses of M/T units in the NPAS4 and control mice is visualized, indicating a small but significant right shift of the empirical cumulative distribution function. Please note that this does not necessarily have to reflect in

visualization of the temporal evolution of the mean z-score response (Fig. 3I) since higher amplitude responses might mask lower amplitude responses.

We clarified this in the results, l. 431: “Specifically, when comparing the genotypes, the odor-inhibited responses had similar amplitudes ($U = 767$, $p = 0.375$, MWU-Test, Fig 3J), similar peak width ($t(85) = 0.3$, $p = 0.797$, unpaired t-test, Fig. 3K), but differed slightly in time to peak response ($U = 630.5$, $p = 0.02$, MWU-Test, Fig. 3L).”

- Line 368 "Thus, in summary, selective developmental deletion of NPAS4 in M/Ts affected bulbar and cortical odor responses in the adult". Cortical responses have not been analysed at this part of the MS.

We thank the reviewer for pointing this out and we removed the cortex statement at this point of the results section.

- Graded shading would be preferred compared to simple binary plots in e.g. Fig 4a /7a right.

We changed the heatmaps in Fig. 4a/7a right to indicate different levels of significance using graded shading.

References cited in the reply

- Abraham NM, Egger V, Shimshek DR, Renden R, Fukunaga I, Sprengel R, Seeburg PH, Klugmann M, Margrie TW, Schaefer AT, Kuner T (2010) Synaptic inhibition in the olfactory bulb accelerates odor discrimination in mice. *Neuron* 65:399–411.
- Alonso M, Lepousez G, Wagner S, Bardy C, Gabellec M-M, Torquet N, Lledo P-M (2012) Activation of adult-born neurons facilitates learning and memory. *Nature Neuroscience* 15:897–904.
- Arevian AC, Kapoor V, Urban NN (2008) Activity-dependent gating of lateral inhibition in the mouse olfactory bulb. *Nat Neurosci* 11:80–87.
- Bloodgood BL, Sharma N, Browne HA, Trepman AZ, Greenberg ME (2013) The activity-dependent transcription factor NPAS4 regulates domain-specific inhibition. *Nature* 503:121–125.
- Brunjes PC, Illig KR, Meyer EA (2005) A field guide to the anterior olfactory nucleus (cortex). *Brain Research Reviews* 50:305–335.
- Chaudhury D, Escanilla O, Linster C (2009) Bulbar Acetylcholine Enhances Neural and Perceptual Odor Discrimination. *J Neurosci* 29:52–60.
- Devore S, Almeida L de, Linster C (2014) Distinct Roles of Bulbar Muscarinic and Nicotinic Receptors in Olfactory Discrimination Learning. *J Neurosci* 34:11244–11260.
- Devore S, Linster C (2012) Noradrenergic and cholinergic modulation of olfactory bulb sensory processing. *Front Behav Neurosci* 6:52.
- Fu J, Guo O, Zhen Z, Zhen J (2020) Essential Functions of the Transcription Factor Npas4 in Neural Circuit Development, Plasticity, and Diseases. *Frontiers in Neuroscience* 14 Available at: <https://www.frontiersin.org/articles/10.3389/fnins.2020.603373> [Accessed August 7, 2023].
- Fujimoto S, Leiwe MN, Aihara S, Sakaguchi R, Muroyama Y, Kobayakawa R, Kobayakawa K, Saito T, Imai T (2023) Activity-dependent local protection and lateral inhibition control synaptic competition in developing mitral cells in mice. *Developmental Cell* 58:1221-1236.e7.
- Imamura K, Mataga N, Mori K (1992) Coding of odor molecules by mitral/tufted cells in rabbit olfactory bulb. I. Aliphatic compounds. *Journal of Neurophysiology* 68:1986–2002.
- Johnson BA, Leon M (2000) Odorant molecular length: One aspect of the olfactory code. *Journal of Comparative Neurology* 426:330–338.
- Kay LM, Laurent G (1999) Odor- and context-dependent modulation of mitral cell activity in behaving rats. *Nat Neurosci* 2:1003–1009.

- Kollo M, Schmaltz A, Abdelhamid M, Fukunaga I, Schaefer AT (2014) "Silent" mitral cells dominate odor responses in the olfactory bulb of awake mice. *Nature Neuroscience* 17:1313–1315.
- Kosaka T, Kosaka K (2012) Further characterization of the juxtglomerular neurons in the mouse main olfactory bulb by transcription factors, Sp8 and Tbx21. *Neuroscience Research* 73:24–31.
- Lepousez G, Lledo P-M (2013) Odor Discrimination Requires Proper Olfactory Fast Oscillations in Awake Mice. *Neuron* 80:1010–1024.
- Lin Y, Bloodgood BL, Hauser JL, Lapan AD, Koon AC, Kim T-K, Hu LS, Malik AN, Greenberg ME (2008) Activity-dependent regulation of inhibitory synapse development by Npas4. *Nature* 455:1198–1204.
- Mandairon N, Stack C, Kiselycznyk C, Linster C (2006) Broad activation of the olfactory bulb produces long-lasting changes in odor perception. *Proceedings of the National Academy of Sciences* 103:13543–13548.
- Margrie TW, Sakmann B, Urban NN (2001) Action potential propagation in mitral cell lateral dendrites is decremental and controls recurrent and lateral inhibition in the mammalian olfactory bulb. *Proc Natl Acad Sci USA* 98:319–324.
- Moreno MM, Linster C, Escanilla O, Sacquet J, Didier A, Mandairon N (2009) Olfactory perceptual learning requires adult neurogenesis. *PNAS* 106:17980–17985.
- Mori K, Mataga N, Imamura K (1992) Differential specificities of single mitral cells in rabbit olfactory bulb for a homologous series of fatty acid odor molecules. *Journal of Neurophysiology* 67:786–789.
- Oettl L-L, Ravi N, Schneider M, Scheller MF, Schneider P, Mitre M, da Silva Gouveia M, Froemke RC, Chao MV, Young WS, Meyer-Lindenberg A, Grinevich V, Shusterman R, Kelsch W (2016) Oxytocin Enhances Social Recognition by Modulating Cortical Control of Early Olfactory Processing. *Neuron* 90:609–621.
- Ravi N, Sanchez-Guardado L, Lois C, Kelsch W (2017) Determination of the connectivity of newborn neurons in mammalian olfactory circuits. *Cell Mol Life Sci* 74:849–867.
- Rinberg D (2006) Sparse Odor Coding in Awake Behaving Mice. *Journal of Neuroscience* 26:8857–8865.
- Rubin BD, Katz LC (1999) Optical Imaging of Odorant Representations in the Mammalian Olfactory Bulb. *Neuron* 23:499–511.
- Sato T, Hirono J, Tonoike M, Takebayashi M (1994) Tuning specificities to aliphatic odorants in mouse olfactory receptor neurons and their local distribution. *Journal of Neurophysiology* 72:2980–2989.

- Shusterman R, Smear MC, Koulakov AA, Rinberg D (2011) Precise olfactory responses tile the sniff cycle. *Nat Neurosci* 14:1039–1044.
- Spiegel I, Mardinly AR, Gabel HW, Bazinet JE, Couch CH, Tzeng CP, Harmin DA, Greenberg ME (2014) *Npas4* regulates excitatory-inhibitory balance within neural circuits through cell-type-specific gene programs. *Cell* 157:1216–1229.
- Wachowiak M, Cohen LB (2001) Representation of Odorants by Receptor Neuron Input to the Mouse Olfactory Bulb. *Neuron* 32:723–735.
- Wolf D, Oettl L-L, Winkelmeier L, Linster C, Kelsch W (2024) Anterior olfactory cortices differentially transform bottom-up odor signals to produce inverse top-down outputs. *J Neurosci*:e0231242024.
- Yokoi M, Mori K, Nakanishi S (1995) Refinement of odor molecule tuning by dendrodendritic synaptic inhibition in the olfactory bulb. *Proc Natl Acad Sci USA* 92:3371–3375.
- Yoshihara S-I, Takahashi H, Nishimura N, Kinoshita M, Asahina R, Kitsuki M, Tatsumi K, Furukawa-Hibi Y, Hirai H, Nagai T, Yamada K, Tsuboi A (2014) *Npas4* regulates *Mdm2* and thus *Dcx* in experience-dependent dendritic spine development of newborn olfactory bulb interneurons. *Cell Rep* 8:843–857.

Dear Dr Kelsch,

Re: JP-RP-2025-288011R1 "**NPAS4 is required in olfactory bulb principal neurons for the expression of lateral inhibition to decode chemically similar odor molecules**" by Wolfgang Kelsch, David Wolf, Lars-Lennart Oettl, Luis Sanchez-Guardado, Christiane Linster, and Carlos Lois

Thank you for submitting your manuscript to The Journal of Physiology. It has been assessed by a Reviewing Editor and by 2 expert referees and we are pleased to tell you that it is acceptable for publication following satisfactory revision.

REVISION CHECKLIST:

Please upload two versions of your manuscript text: one with all relevant changes highlighted and one clean version with no changes tracked. The manuscript file should include all tables and figure legends, but each figure/graph should be uploaded as separate, high-resolution files. The journal is now integrated with Wiley's Image Checking service. For further details, see: <https://www.wiley.com/en-us/network/publishing/research-publishing/trending-stories/upholding-image-integrity-wileys->

image-screening-service

We look forward to receiving your revised submission.

Yours sincerely,

Nathan Schoppa
Senior Editor
The Journal of Physiology

EDITOR COMMENTS

Thank you for sending your revised manuscript to Journal of Physiology. We are pleased to inform you that the referees felt that your revisions have addressed most of their prior concerns, for example in the readability of the manuscript. However, a few minor concerns do remain. One more round of revisions and evaluation by at least one of the referees and myself as reviewing editor will be required prior to final acceptance of the study. The remaining points from the referee and reviewing editor include:

1. As indicated by referee 2, the authors should retest for chemical similarity in the AON using the current data to substantiate their claim that similarity coding is not a feature in AON of WT mice.
2. The authors should address all remaining concerns about wording from referee 2.
3. In response to a prior request that the authors provide literature-supported speculation about the mechanisms linking a change in expression of a transcription factor NPAS4 to changes in the neural responses, the authors have now added significant text in the Introduction discussing how NPAS4 has been shown in the neocortex to alter inhibitory synaptic connectivity onto excitatory cells. Moreover, they use the evidence in neocortex to motivate their study: "We therefore probed this transcription factor to explore the role of inhibitory synaptic wiring on odor representations in M/T and the downstream AON in adult mice." This seems inappropriate, in the absence of evidence that NPAS4 alters inhibitory synaptic connections in the olfactory system. The authors should remove this text that motivates their study from the Introduction and instead spend time in the Discussion discussing potential mechanisms such as altered inhibitory synaptic connections in olfactory circuits that would need to be tested in future studies.
4. Along a similar line as Point 3, the authors need to adjust their wording in their Key Points that is based on the assumption that NPAS4 has the same effect on inhibitory synapses in MOB as seen in the neocortex. The second Key Point states: "We tested this assumption by selective mutation of a transcription factor for activity-dependent inhibitory synaptic wiring." It is in fact not known that this transcription factor is for activity-dependent inhibitory synaptic wiring in MOB.
5. The authors should remove from the title the referencing of "expression of lateral inhibition". The authors observe effects of altering expression of NPAS-4 on odor-evoked inhibition, but it is not clear that this is lateral inhibition, at least as it is generally understood.

6. The Introduction states: "Mitral and the less numerous tufted cells (hereafter collectively referred to as M/Ts) of the MOB are glutamatergic projection neurons..." At least most classical studies suggest that tufted cells are more numerous than mitral cells. Unless the authors know of other evidence suggesting that mitral cells are more numerous, the authors should alter this wording.

REFeree COMMENTS

Referee #1:

The authors have addressed all the questions and concerns raised in the previous round of reviews. I have no further comments or concerns.

Referee #2:

This reviewer thanks the authors for clarifying most points. The readability of the manuscript improved significantly. Some minor remaining points should be addressed.:

- Line 399 "selectively" should be removed
- In line 540 the authors state that similarity coding is not a feature in AON of WT mice (and cite their work). In line 624/625 of the discussion, they state that "chemical similarity was discarded for aldehydes". This reads a bit like a circular argument. The authors should retest for chemical similarity in the AON using the current MS data to substantiate this claim.
- Line 549 "cortical neurons" instead of M/T cells?
- Line 598, please add more references to (Wolf et al)
- Line 632, where can the "less broad responses to still recruited neurons" be seen? Isn't this a contradiction to Fig 7c?
- While refusing to be the reviewer to ask for additional experiments, olfactory perception testing would significantly increase the impact of the whole story.

END OF COMMENTS

Dear Dr. Schoppa,

We hereby submit the revised ms. with all points addressed:

EDITOR COMMENTS

Thank you for sending your revised manuscript to Journal of Physiology. We are pleased to inform you that the referees felt that your revisions have addressed most of their prior concerns, for example in the readability of the manuscript. However, a few minor concerns do remain. One more round of revisions and evaluation by at least one of the referees and myself as reviewing editor will be required prior to final acceptance of the study. The remaining points from the referee and reviewing editor include:

We thank you and the Referees for their careful reevaluation and constructive feedback; acknowledging that most concerns have been addressed, we provide point-by-point responses and further revisions to resolve the remaining minor issues.

We have consulted The Journal's Rigour and Reproducibility requirements, the Journal's policy for the Methods section and statistics policy and accordingly:

- updated Figs. 2C1-2, 3E-F, 3J-K, 4C-D, 5D1-2, 6E-F, 6J-K and 7D and the corresponding legends to now illustrate the distribution of the data using box plots
- added a section 'Ethical approval' to the methods section
- added detailed information on dosage of analgesia and anesthesia procedures
- reported all p-values to three significant figures if available

The line numbers indicated in this reply refer to the numbers of the attached manuscript .pdf with highlighted changes in red.

1. As indicated by referee 2, the authors should retest for chemical similarity in the AON using the current data to substantiate their claim that similarity coding is not a feature in AON of WT mice.

We added this information in the text of the result section (L. 544): **"We also revisited the coding of chemical similarity in the AON (Fig. 7B). As previously observed in an analysis of these recordings (Wolf et al., 2024b), we neither observed significant chemical similarity coding in control mice nor in Δ NPAS4^{M/T} mice (main effect of c-chain length: $F(4,1330) = 1.443$, $p = 0.2176$, main effect of genotype: $F(1,1330) = 90.35$, $p < 0.0001$, 2-way repeated-measures ANOVA, Šidák multiple comparison tests were all not significant, see Fig. 7B, right)."**

2. The authors should address all remaining concerns about wording from referee 2.

We addressed all concerns regarding wording as specified below in response to referee 2.

3. In response to a prior request that the authors provide literature-supported speculation about the mechanisms linking a change in expression of a transcription factor NPAS4 to

changes in the neural responses, the authors have now added significant text in the Introduction discussing how NPAS4 has been shown in the neocortex to alter inhibitory synaptic connectivity onto excitatory cells. Moreover, they use the evidence in neocortex to motivate their study: "We therefore probed this transcription factor to explore the role of inhibitory synaptic wiring on odor representations in M/T and the downstream AON in adult mice." This seems inappropriate, in the absence of evidence that NPAS4 alters inhibitory synaptic connections in the olfactory system. The authors should remove this text that motivates their study from the Introduction and instead spend time in the Discussion discussing potential mechanisms such as altered inhibitory synaptic connections in olfactory circuits that would need to be tested in future studies.

We revised the motivating sentence to more accurately reflect the previous literature and the scope of our study, L. 126: "~~Accordingly, we examined how developmental deletion of NPAS4 in M/Ts affects odor-evoked representations in adult mice. We therefore probed this transcription factor to explore the role of inhibitory synaptic wiring on odor representations in M/T and the downstream AON in adult mice.~~"

And we added to the discussion, L. 608: "Future studies may examine the synaptic changes emerging from deletion of NPAS4 in olfactory bulb projection neurons."

4. Along a similar line as Point 3, the authors need to adjust their wording in their Key Points that is based on the assumption that NPAS4 has the same effect on inhibitory synapses in MOB as seen in the neocortex. The second Key Point states: "We tested this assumption by selective mutation of a transcription factor for activity-dependent inhibitory synaptic wiring." It is in fact not known that this transcription factor is for activity-dependent inhibitory synaptic wiring in MOB.

We revised the Key Points: "~~2. We tested this assumption by selective mutation of a transcription factor for activity-dependent inhibitory synaptic wiring.~~—We tested how conditional deletion of the activity-dependent transcription factor NPAS4 in mitral/tufted cells affects bulbar and cortical odor representations."

5. The authors should remove from the title the referencing of "expression of lateral inhibition". The authors observe effects of altering expression of NPAS-4 on odor-evoked inhibition, but it is not clear that this is lateral inhibition, at least as it is generally understood.

We changed the title of the study to "~~Deletion of NPAS4 in olfactory bulb principal neurons alters E/I balance and impairs decoding of chemically similar odor molecules~~".

6. The Introduction states: "Mitral and the less numerous tufted cells (hereafter collectively referred to as M/Ts) of the MOB are glutamatergic projection neurons..." At least most classical studies suggest that tufted cells are more numerous than mitral cells. Unless the authors know of other evidence suggesting that mitral cells are more numerous, the authors should alter this wording.

We thank the editor for pointing this out and adapted the wording to now state in L. 102: "Mitral and ~~the less numerous~~ tufted cells (hereafter collectively referred to as M/Ts) of the MOB are

glutamatergic projection neurons that have one apical dendrite preferentially targeting one glomerulus, and multiple lateral dendrites.”

REFEREE COMMENTS

Referee #1:

The authors have addressed all the questions and concerns raised in the previous round of reviews. I have no further comments or concerns.

We thank the reviewer for the re-evaluation and assessment.

Referee #2:

This reviewer thanks the authors for clarifying most points. The readability of the manuscript improved significantly. Some minor remaining points should be addressed.:

We appreciate the reviewer’s reevaluation and addressed all remaining points:

- Line 399 "selectively" should be removed

We removed “selectively” as indicated by the reviewer.

- In line 540 the authors state that similarity coding is not a feature in AON of WT mice (and cite their work). In line 624/625 of the discussion, they state that "chemical similarity was discarded for aldehydes". This reads a bit like a circular argument. The authors should retest for chemical similarity in the AON using the current MS data to substantiate this claim.

We added this information in the text of the result section (L. 544): “We also revisited the coding of chemical similarity in the AON (Fig. 7B). As previously observed in an analysis of these recordings (Wolf et al., 2024b), we neither observed significant chemical similarity coding in control mice nor in Δ NPAS4^{M/T} mice (main effect of c-chain length: $F(4,1330) = 1.443$, $p = 0.2176$, main effect of genotype: $F(1,1330) = 90.35$, $p < 0.0001$, 2-way repeated-measures ANOVA, Šídák multiple comparison tests were all not significant, see Fig. 7B, right).”

Also, we removed the following sentences from the discussion as they are not strictly needed for the discussion and the other points are made elsewhere in the discussion: “~~The odor coding in the AON resembles that of other olfactory cortices (Stettler and Axel, 2009; Bolding and Franks, 2017). Thus, while the MOB is dominated by more complex response patterns emerging from lateral inhibition and temporal coding (e.g. this study, Sato et al., 1994; Yokoi et al., 1995; Uchida et al., 2014), the more prominent odor-excited rate code in the AON generally discriminated the different aldehydes reliably in control mice, but the additional information of chemical similarity was discarded for aldehydes.~~”

- Line 549 "cortical neurons" instead of M/T cells?

We removed "and the input from M/T cells" from line 553/554 for clarification.

- Line 598, please add more references to (Wolf et al)

We added two references to line 596/597, supporting that chemically dissimilar molecules are represented more differently in the olfactory bulb (Johnson and Leon, 2000; Wachowiak and Cohen, 2001).

- Line 632, where can the "less broad responses to still recruited neurons" be seen? Isn't this a contradiction to Fig 7c?

We thank the reviewer for pointing this out and now state (L.629): "The ability to decode was also not compensated by increased sparseness, namely the lower percentage of recruited cortical neurons to any odor in the mutant mice."

- While refusing to be the reviewer to ask for additional experiments, olfactory perception testing would significantly increase the impact of the whole story.

We appreciate not requesting additional experiments. Given the time and resource requirements for a rigorous behavioral evaluation, we have deferred these experiments to a subsequent study, as stated in the Discussion.

Dear Dr Kelsch,

Re: JP-RP-2025-288011R2 "**Deletion of NPAS4 in olfactory bulb principal neurons alters E/I balance and impairs decoding of chemically similar odor molecules**" by Wolfgang Kelsch, David Wolf, Lars-Lennart Oettl, Luis Sanchez-Guardado, Christiane Linster, and Carlos Lois

Thank you for submitting your manuscript to The Journal of Physiology. It has been assessed by a Reviewing Editor and by 1 expert referees and we are pleased to tell you that it is acceptable for publication following satisfactory revision.

REVISION CHECKLIST:

Please upload two versions of your manuscript text: one with all relevant changes highlighted and one clean version with no changes tracked. The manuscript file should include all tables and figure legends, but each figure/graph should be uploaded as separate, high-resolution files. The journal is now integrated with Wiley's Image Checking service. For further details, see: <https://www.wiley.com/en-us/network/publishing/research-publishing/trending-stories/upholding-image-integrity-wileys->

image-screening-service

We look forward to receiving your revised submission.

Yours sincerely,

Nathan Schoppa
Senior Editor
The Journal of Physiology

REQUIRED ITEMS

- 1) - Author photo and profile should be uploaded and clearly labelled together in one Word document with the revised version of the manuscript. See Information for Authors for further details.
- 2) - You must start the Methods section with a paragraph headed Ethical approval (https://jp.msubmit.net/cgi-bin/main.plex?form_type=display_requirements#methods).

Please include the ethics committee reference number and the route of anaesthesia.

Research must comply with The Journal's policies regarding animal experiments (<https://physoc.onlinelibrary.wiley.com/hub/animal-experiments>) and adherence to these policies must be stated in the manuscript.

Authors should confirm in their Methods section that their experiments were carried out according to the guidelines laid down by their institution's animal welfare committee, including an ethics approval reference number. The Methods section must contain a statement about access to food, water and housing, details of the anaesthetic regime: anaesthetic used, dose and route of administration, and method of killing the experimental animals.

- 3) - Your manuscript must include a complete Additional Information section, including competing interests; funding; author contributions and acknowledgements.

- 4) - Papers must comply with the Statistics Policy: https://jp.msubmit.net/cgi-bin/main.plex?form_type=display_requirements#statistics.

In summary:

- If $n \leq 30$, all data points must be plotted in the figure in a way that reveals their range and distribution. A bar graph with data points overlaid, a box and whisker plot or a violin plot (preferably with data points included) are acceptable formats.
- If $n > 30$, then the entire raw dataset must be made available either as supporting information, or hosted on a not-for-profit repository, e.g. FigShare, with access details provided in the manuscript.
- 'n' clearly defined (e.g. x cells from y slices in z animals) in the Methods. Authors should be mindful of pseudoreplication.
- All relevant 'n' values must be clearly stated in the main text, figures and tables.
- The most appropriate summary statistic (e.g. mean or median and standard deviation) must be used. Standard Error of the Mean (SEM) alone is not permitted.
- Exact p values must be stated. Authors must not use 'greater than' or 'less than'. Exact p values must be stated to three

significant figures even when 'no statistical significance' is claimed.

EDITOR COMMENTS

Senior Editor:

Thank you for submitting your revised manuscript. The authors have addressed nearly all of the prior concerns that were raised in a satisfactory manner. However, referee 2 has two remaining minor points. He/she states that one request was not addressed and, also, that the new analysis that was added to retest for chemical similarity in AON raised a new issue. I also am confused about the conclusion that similarity coding is not a feature in AON of WT or Δ NPAS4M/T mice, but that there appeared to be a significant effect of genotype in the analysis. The authors should clarify.

REFEREE COMMENTS

Referee #2:

- Line 399 "selectively" should be removed

We removed "selectively" as indicated by the reviewer.

--This has not been corrected (L 405).

- In line 540 the authors state that similarity coding is not a feature in AON of WT mice (and cite their work). In line 624/625 of the discussion, they state that "chemical similarity was discarded for aldehydes". This reads a bit like a circular argument. The authors should retest for chemical similarity in the AON using the current MS data to substantiate this claim.

We added this information in the text of the result section (L. 544): "We also revisited the coding of chemical similarity in the AON (Fig. 7B). As previously observed in an analysis of these recordings (Wolf et al., 2024b), we neither observed significant chemical similarity coding in control mice nor in Δ NPAS4M/T mice (main effect of c-chain length: $F(4,1330) = 1.443$, $p = 0.2176$, main effect of genotype: $F(1,1330) = 90.35$, $p < 0.0001$, 2-way repeated-measures ANOVA, Šídák multiple comparison tests were all not significant, see Fig. 7B, right)."

--Thank you for adding the additional analysis. However, the result seems confusing. Similarity coding is not a feature in AON of WT or Δ NPAS4M/T mice, but there is a significant effect of genotype? Please explain.

END OF COMMENTS

EDITOR COMMENTS

Senior Editor:

Thank you for submitting your revised manuscript. The authors have addressed nearly all of the prior concerns that were raised in a satisfactory manner. However, referee 2 has two remaining minor points. He/she states that one request was not addressed and, also, that the new analysis that was added to retest for chemical similarity in AON raised a new issue. I also am confused about the conclusion that similarity coding is not a feature in AON of WT or Δ NPAS4M/T mice, but that there appeared to be a significant effect of genotype in the analysis. The authors should clarify.

We thank the Editor and Referee for their reevaluation and address the two remaining points below.

We have also checked that the revised manuscript complies with the required items stated in the Decision letter. Specifically, we

- 1) Uploaded the author photo and profile in one Word document.
- 2) Added the ethics committee reference number in the 'Ethical Approval' Section of the Methods and confirm that the Methods contain a statement about access to food, water and housing ('Animals and Husbandry'), details of the anaesthetic regime: anaesthetic used, dose and route of administration, and method of killing the experimental animals (specified in 'Immunohistochemistry', 'In-vivo electrophysiology' and 'Histological confirmation of recording site').
- 3) Include a complete Additional Information Section.
- 4) We made the following changes and now confirm that the paper fully complies with the Statistics Policy:
 - a. We provide a Supporting Information file containing the raw data
 - b. Also in the Supporting Information file, we provide p-value tables where there are many post-hoc comparisons (e.g., Fig. 4A, D).
 - c. We updated the figure panels Fig. 3D, I and Fig. 6D, I to show mean \pm standard deviation instead of mean \pm SEM.

The line numbers indicated in this reply refer to the numbers of the attached manuscript .pdf with highlighted changes in red.

REFEREE COMMENTS

Referee #2:

- Line 399 "selectively" should be removed

We removed "selectively" as indicated by the reviewer.

--This has not been corrected (L 405).

We apologize for this mistake. In response to the reviewer's earlier comment, we deleted the word "selective" from the sentence below by mistake, due to confusion between different manuscript versions with a line-numbering offset between the PDF from the submission portal and the Word file.

"In the AON, only axons and their terminal buttons expressed tdTomato (Fig. 1F2), consistent with selective expression of Tbet-Cre in the MOB, but not AON."

We now also removed the other "selectively" as requested by the reviewer (l. 377):

"The gene of interest was deleted conditionally by Cre-recombination under the Tbet promoter (also known as *Tbx21*, Haddad et al., 2013) that is selectively expressed in M/Ts approximately from embryonic day 14 on (Faedo et al., 2002; Mizuguchi et al., 2012)."

- In line 540 the authors state that similarity coding is not a feature in AON of WT mice (and cite their work). In line 624/625 of the discussion, they state that "chemical similarity was discarded for aldehydes". This reads a bit like a circular argument. The authors should retest for chemical similarity in the AON using the current MS data to substantiate this claim.

We added this information in the text of the result section (L. 544): "We also revisited the coding of chemical similarity in the AON (Fig. 7B). As previously observed in an analysis of these recordings (Wolf et al., 2024b), we neither observed significant chemical similarity coding in control mice nor in Δ NPAS4M/T mice (main effect of c-chain length: $F(4,1330) = 1.443$, $p = 0.2176$, main effect of genotype: $F(1,1330) = 90.35$, $p < 0.0001$, 2-way repeated-measures ANOVA, Šídák multiple comparison tests were all not significant, see Fig. 7B, right)."

--Thank you for adding the additional analysis. However, the result seems confusing. Similarity coding is not a feature in AON of WT or Δ NPAS4M/T mice, but there is a significant effect of genotype? Please explain.

We thank the reviewer for requesting this clarification. We now explained the test results better: The main effect of genotype in the two-way ANOVA reflects the impact of NPAS4 deletion on overall response differentiation and the increased uniformity of odor responses (cf. Fig. 7A, D, E). The conclusion that chemical similarity is not a coding feature in the AON, comes from the testing of the main effect of carbon-chain length and the posthoc tests that both were not significant.

We revised it accordingly, l. 531:

“We also revisited the coding of chemical similarity in the AON (Fig. 7B). As previously observed in an analysis of these recordings (Wolf et al., 2024b), **we found no evidence for chemical similarity coding in the AON. As expected, the two-way repeated-measures ANOVA revealed a significant main effect of genotype ($F(1,1330) = 90.35$, $p < 0.0001$), which matches the overall reduction and uniformity of odor responses caused by NPAS4 deletion (cf. Fig. 7A). The main effect of carbon-chain length was however not significant ($F(4,1330) = 1.443$, $p = 0.2176$), and all Šídák-corrected post-hoc tests comparing responses across carbon-chain lengths were non-significant (see Fig. 7B, right), supporting that chemical similarity coding is not significantly expressed in the AON in either control or Δ NPAS4^{M/T} mice.”**

Dear Dr Kelsch,

Re: JP-RP-2025-288011R3 "**Deletion of NPAS4 in olfactory bulb principal neurons alters E/I balance and impairs decoding of chemically similar odor molecules**" by Wolfgang Kelsch, David Wolf, Lars-Lennart Oettl, Luis Sanchez-Guardado, Christiane Linster, and Carlos Lois

We are pleased to tell you that your paper has been accepted for publication in The Journal of Physiology.

Yours sincerely,

Nathan Schoppa
Senior Editor
The Journal of Physiology

IMPORTANT POINTS TO NOTE FOLLOWING ACCEPTANCE OF YOUR PAPER:

- **IMPORTANT NOTICE ABOUT OPEN ACCESS:** To assist authors whose funding agencies mandate immediate public access to published research findings, The Journal of Physiology allows authors to pay an Open Access (OA) fee to have their papers made freely available immediately on publication.

- You can help your research get the attention it deserves! Check out Wiley's free Promotion Guide for best-practice recommendations for promoting your work at: www.wileyauthors.com/eoo/guide. You can learn more about Wiley Editing Services which offers professional video, design, and writing services to create shareable video abstracts, infographics, conference posters, lay summaries, and research news stories for your research at: www.wileyauthors.com/eoo/promotion.

- If you would like to receive our 'Research Roundup', a monthly newsletter highlighting the cutting-edge research published in The Physiological Society's family of journals (The Journal of Physiology, Experimental Physiology, Physiological Reports, The Journal of Nutritional Physiology and The Journal of Precision Medicine: Health and Disease), please click this link, fill in your name and email address and select 'Research Roundup': <https://www.physoc.org/journals-and-media/membernews>

EDITOR COMMENTS

Reviewing Editor:

Thank you for addressing the final remaining points that were raised about your manuscript. Your work is now acceptable for publication!

Senior Editor:

Thank you for addressing the final remaining points that were raised. Your work is now acceptable for publication.